# Deep Fuzzy Multi-view Learning for Reliable Classification

**Siyuan Duan**[1]   **Yuan Sun**[1]   **Dezhong Peng**[1 2 3]   **Guiduo Duan**[4]   **Xi Peng**[1]   **Peng Hu**[1]

## Abstract

Multi-view learning methods primarily focus on enhancing decision accuracy but often neglect the uncertainty arising from the intrinsic drawbacks of data, such as noise, conflicts, etc. To address this issue, several trusted multi-view learning approaches based on the Evidential Theory have been proposed to capture uncertainty in multi-view data. However, their performance is highly sensitive to conflicting views, and their uncertainty estimates, which depend on the total evidence and the number of categories, often underestimate uncertainty for conflicting multi-view instances due to the neglect of inherent conflicts between belief masses. To accurately classify conflicting multi-view instances and precisely estimate their intrinsic uncertainty, we present a novel Deep Fuzzy Multi-View Learning (**FUML**) method. Specifically, FUML leverages Fuzzy Set Theory to model the outputs of a classification neural network as fuzzy memberships, incorporating both possibility and necessity measures to quantify category credibility. A tailored loss function is then proposed to optimize the category credibility. To further enhance uncertainty estimation, we propose an entropy-based uncertainty estimation method leveraging category credibility. Additionally, we develop a Dual Reliable Multi-view Fusion (DRF) strategy that accounts for both view-specific uncertainty and inter-view conflict to mitigate the influence of conflicting views in multi-view fusion. Extensive experiments demonstrate that our FUML achieves state-of-the-art performance in terms of both accuracy and reliability.

---

[1]College of Computer Science, Sichuan University, Chengdu, China. [2] Sichuan National Innovation New Vision UHD Video Technology Co., Ltd, Chengdu, China. [3]Tianfu Jincheng Laboratory, Chengdu, China. [4] School of Computer Science and Engineering, University of Electronic Science and Technology of China, Chengdu, China. Correspondence to: Peng Hu <penghu.ml@gmail.com>.

*Proceedings of the 42st International Conference on Machine Learning*, Vancouver, Canada. PMLR 267, 2025. Copyright 2025 by the author(s).

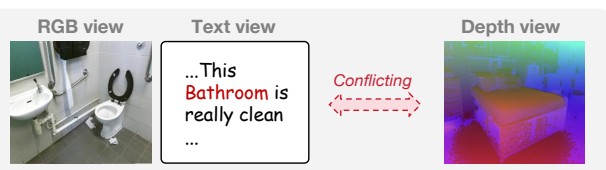

(a) Visualization of the conflicting multi-view instance

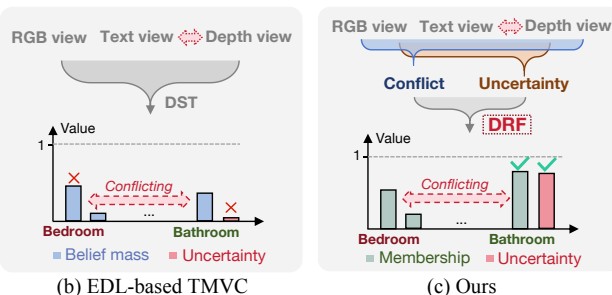

(b) EDL-based TMVC          (c) Ours

*Figure 1.* (a) Visualization of the conflicting multi-view instance: the depth view is related to the "Bedroom" category, while the other views show conflicting information, such as "Bathroom." (b) EDL-based TMVC methods are sensitive to such conflicting multi-view instances. On one hand, because they neglect the global conflict between views in multi-view fusion, classification errors are often made. On the other hand, their uncertainty estimation is only related to the total evidence and the number of categories. For conflicting multi-view instances, as long as the total evidence is large, the uncertainty is seriously underestimated. (c) In our method, both global conflict and uncertainty are considered during fusion, allowing the conflicting multi-view instances to be classified correctly. Additionally, this method can estimate decision uncertainty more accurately.

## 1. Introduction

Multi-view/modal data encapsulates comprehensive information from various modalities, sources, and other perspectives (Yan et al., 2021; Qin et al., 2024; He et al., 2024). Multi-view classification (MVC) aims to utilize the consistency and complementary nature of these data to achieve more accurate classification. With the explosive growth of multi-view data in fields such as video surveillance (Wang et al., 2024a), medical detection (Yang et al., 2024; Nasarian et al., 2024), and autonomous driving (Hong et al., 2024), MVC has garnered great attention from both academia and industry in recent years. Despite the promising performance of existing MVC methods (Yang et al., 2019; Han et al., 2022a; Lin et al., 2022; Mittal et al., 2022; Zhang et al.,

2023), these approaches predominantly prioritize classification accuracy while often neglecting decision uncertainty. This could lead to unreliable decisions, limiting their applicability in reliability-critical scenarios.

To address this limitation, a series of trusted multi-view classification (TMVC) methods (Geng et al., 2021; Han et al., 2020; Liu et al., 2022; Xie et al., 2023; Liu et al., 2023; Xu et al., 2024b) based on Evidential Deep Learning (EDL) (Sensoy et al., 2018; Bao et al., 2021) have been proposed to estimate the uncertainty. These methods provide classification predictions alongside the corresponding uncertainty (inverted to reliability). However, they typically assume strict alignment across different views, which is often unrealistic due to noise, misalignment, or conflicting information in real-world scenarios. More intuitively, a case of multi-view user-generated content is illustrated in Figure 1 (a), where the RGB, text, and depth views exhibit conflicting categorical information. Such conflicts pose significant challenges to EDL-based TMVC methods as shown in Figure 1 (b), remarkably degrading their performance. Specifically, these methods face two key issues: i) They heavily rely on Dempster-Shafer theory (DST) (Shafer, 1992) for multi-view fusion, which fails to account for global conflicts among views overly emphasizes dominant evidence (Xiao, 2019b; Shang et al., 2021), often leading to misclassification. ii) Their uncertainty estimation relies solely on total evidence and the number of categories, neglecting inherent conflicts between belief masses, leading to inaccurate uncertainty quantification for conflicting multi-view instances.

To address the aforementioned problems, this paper presents a novel framework, Deep Fuzzy Multi-View Learning (**FUML**), grounded in Fuzzy Set Theory (Zadeh, 1965), which provides more precise decisions along with the corresponding accurate uncertainty estimates. Fuzzy Set Theory manages inherent uncertainty and fuzziness in data by introducing gradual membership between 0 and 1, allowing a sample to belong to multiple categories to varying degrees, thereby enabling effective uncertainty quantization and modeling. Based on this principle, as shown in Figure 2, we model the outputs of the classification network as fuzzy memberships corresponding to each category, representing the extent to which a sample belongs to each category. However, memberships alone provide only a possibility measure, i.e., the likelihood of belonging to a category, without indicating the between-class relationship (called necessity) that the sample does not belong to other categories. To integrate both aspects, we introduce category credibility, optimized via the proposed category credibility learning loss. Furthermore, we propose an entropy-based uncertainty estimation method that leverages category credibility to enhance uncertainty quantification. To mitigate the impact of conflicting views in multi-view fusion, we develop a Dual-reliable Multi-view Fusion (DRF) strategy, which considers both

view-specific decision uncertainty and inter-view conflicts. Unlike existing uncertainty-aware fusion techniques in EDL-based TMVC that operate sequentially, our DRF employs a global one-time fusion strategy that reduces the influence of high-uncertainty and high-conflict views, ensuring more robust multi-view classification. The main contributions of this work are summarized as follows:

- We reveal and address the conflict sensitivity issue in existing EDL-based TMVC methods, proposing FUML, a novel framework based on Fuzzy Set Theory for enhancing classification and uncertainty estimation.

- We develop a Dual-reliable Multi-view Fusion (DRF) strategy that effectively reduces the impact of conflicting views, embracing more robust multi-view classification.

- We propose an entropy-based uncertainty qualification technique, enabling more accurate uncertainty estimation for conflicting multi-view instances.

- We conduct extensive experiments comparing our FUML against 13 state-of-the-art MVC baselines on eight widely-used benchmarks, demonstrating superior accuracy, reliability, and robustness.

## 2. Related Work

**Multi-view Learning.** Studies have demonstrated that multi-view learning (MVL) significantly enhances performance across various tasks. Among them, the CCA-based multi-view methods are representative (Chaudhuri et al., 2009; Rupnik & Shawe-Taylor, 2010). With the advancements in deep learning (Yan et al., 2020), some deep MVL methods (Han et al., 2022a; Lin et al., 2022; Zhang et al., 2023; Cao et al., 2024; Qu et al., 2024; Wang et al., 2024b; Chen et al., 2024; Bi & Dornaika, 2024; Xu et al., 2025) emerged. However, most of them focus on improving accuracy, ignoring reliability, limiting their applicability in reliability-critical domains. To achieve reliable decisions, a range of trusted muli-view classification (TMVC) methods (Geng et al., 2021; Han et al., 2020; Liu et al., 2022; 2023; Xie et al., 2023; Liu et al., 2023; Xu et al., 2024a;b; Wang et al., 2024c; Yue et al., 2025; Wang et al., 2025) based on Evidential Deep Learning (EDL) (Sensoy et al., 2018; Gao et al., 2024) and Dempster-Shafer theory (DST) (Shafer, 1992) are proposed. Among them, Trusted Multi-view Classification (TMC) (Han et al., 2020) and Enhanced TMC (ETMC) (Han et al., 2022b) assume that multi-view data is complete; they dynamically integrate different views at the evidence level. However, the arbitrary view missing is widely present in practice. To solve this, Uncertainty-induced Incomplete Multi-View Data Classification (UIMC) (Xie et al., 2023) is proposed. Neverthe-

less, UIMC assumes views are strictly aligned, while multi-view data often contains low-quality conflicting instances. To address this, Evidential Conflictive Multiview Learning (ECML) (Xu et al., 2024a) designs a conflict opinion aggregation strategy and achieves reliable results for conflicting instances. However, ECML overly relies on the latter combined views, making its final decision vulnerable to conflicting views. Given this, we suggest jumping out of the Evidence Theory, and re-examining TMVC based on the Fuzzy Set Theory (Zadeh, 1965; Liu & Liu, 2010) to achieve a more accurate and robust TMVC.

**Uncertainty-aware Deep Learning.** Although deep learning has achieved great success in many tasks, it is difficult to provide reliable uncertainty estimates, which is crucial for reliable models (Wen et al., 2023; Chen et al., 2023). To solve this, early works endowed Deep Neural Networks (DNNs) with uncertainty by using distributions instead of deterministic weight parameters (Gal & Ghahramani, 2015; Molchanov et al., 2017; Lakshminarayanan et al., 2017), but they often suffer from high computational costs. The recently proposed test-time augmentation (Lyzhov et al., 2020) estimates uncertainty at test time, but it still needs multiple inferences. In contrast, Evidential Deep Learning (EDL) (Sensoy et al., 2018; Qin et al., 2022; Li et al., 2025) directly infers uncertainty from network outputs. Recently, researchers have extended EDL to the field of multi-view learning and pioneered a series of methods (Geng et al., 2021; Han et al., 2020; Liu et al., 2022; Xie et al., 2023; Liu et al., 2023; Xu et al., 2024a;b). Although these methods achieve promising uncertainty estimates, their uncertainty depends only on the number of categories and the total evidence, and the uncertainty of conflicting multi-view instance is often underestimated. In this paper, we draw on the Fuzzy Set Theory (Zadeh, 1965; Liu & Liu, 2010), which provides a more nuanced perspective that combines possibility and necessity measures to capture uncertainty more accurately.

## 3. The Proposed Method

### 3.1. Problem Definition

For a clear presentation, we first introduce the following notations. Given $N$ training inputs $\{\mathcal{X}_n\}_{n=1}^N$ with $V$ views, i.e., $\mathcal{X} = \{\mathbf{x}^v\}_{v=1}^V$, and the corresponding labels $\{\mathbf{y}_n\}_{n=1}^N$. The goal of trusted multi-view classification is to learn a model $f : \{\mathbf{x}^v\}_{v=1}^V \to \mathbf{y}$ that accurately predicts the label for an unseen sample by effectively integrating information from all available views and provide the corresponding decision uncertainty. The main challenge lies in utilizing the consistent and complementary information from each view while managing conflicting views, ultimately improving overall classification performance and providing accurate uncertainty estimation.

### 3.2. Deep Fuzzy Multi-view Learning

#### 3.2.1. CATEGORY CREDIBILITY MODELING

The core idea of Fuzzy Set Theory is to allow samples to belong to a set to a certain degree, rather than either absolutely belonging or not belonging (Zadeh, 1965). Based on this, fuzzy systems can effectively handle the uncertainties and ambiguities inherent in real-world data (Das et al., 2020; Wu et al., 2025). According to Fuzzy Set Theory, membership quantifies the degree to which a sample belongs to a fuzzy set. Similarly, the output probabilities of a classification network, ranging from 0 to 1, represent the possibility of a sample belonging to each category, with higher values indicating a greater possibility of classification into that category. This parallel enables us to model the classifier's prediction for a category as a membership for that category. Therefore, for a given sample $\mathbf{x}_i^v$, the memberships across all categories can be expressed as $m_{i1}^v, m_{i2}^v, \ldots, m_{iK}^v$, where $K$ represents the total number of categories.

Nevertheless, the memberships only provide the **possibility measure** which represents the likelihood of belonging to a category, not the **necessity measure**—the certainty that the sample does not belong to other categories (Liu & Liu, 2010; Duan et al., 2025). To complement membership by quantifying these certainties, we introduce the concept of necessity:

$$e_{ik}^v = 1 - \max\{m_{il}^v \mid l \neq k\}, \ k = 1, ..., K, \quad (1)$$

where $\max\{m_{il}^v \mid l \neq k\}$ is the highest membership for the other categories $\{l\}_{l \neq k}$. By integrating both possibility measure and necessity measure, we can obtain more comprehensive information. Therefore, we define the category credibility as the arithmetic mean of the possibility measure and the necessity measure:

**Definition 3.1.** Let $\mathbf{m}_i^v = [m_{i1}^v, m_{i2}^v, ..., m_{iK}^v]$ be the vector of memberships of the $i$-th sample in $v$-th view, and $\forall m_{ik}^v \in [0, 1], \ k = 1, 2, ..., K$. Then, the category credibility of the $i$-th sample to the $k$-th category is defined by

$$c_{ik}^v = \frac{1}{2}(m_{ik}^v + 1 - \max\{m_{il}^v \mid l \neq k\}), \ k = 1, 2, ..., K,$$
$$(2)$$

which can be arranged as $\mathbf{c}_i^v = [c_{i1}^v, c_{i2}^v, ..., c_{iK}^v] \in \mathbb{R}^K$.

#### 3.2.2. CATEGORY CREDIBILITY LEARNING

To map the logits of a neural network as memberships, first, $L_p$-normalization is applied to the logits to limit the value in the range of [-1,1]. Subsequently, an activation function (i.e., ReLU) is used to yield output values in the range of [0,1]. These values can be modeled as memberships for the corresponding category. To be specific, the calculation formula is as follows:

$$\mathbf{m}_i^v = \text{ReLU}\left(\frac{\mathbf{a}_i^v}{||\mathbf{a}_i^v||_p}\right), \quad (3)$$

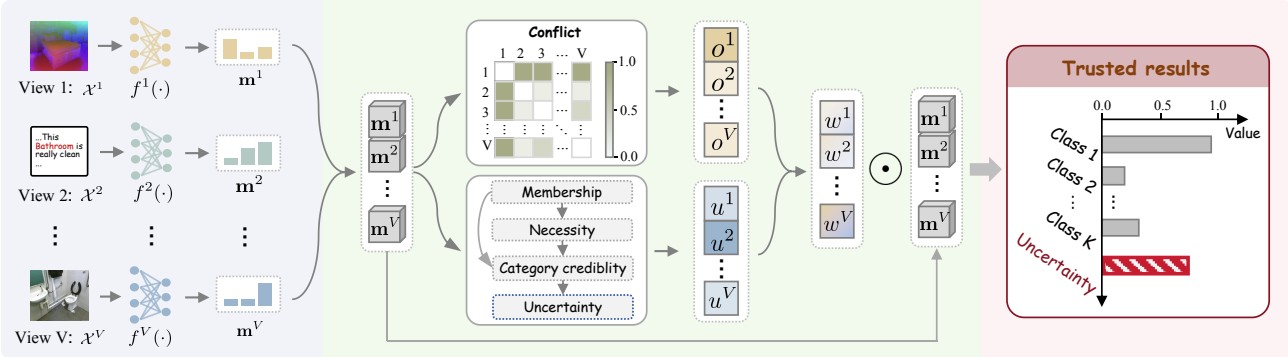

*Figure 2.* Illustration of FUML. Firstly, view-specific DNNs ($\{f^v(\cdot)\}_{v=1}^V$) collect memberships ($\{\mathbf{m}^v\}_{v=1}^V$) from multi-view instances ($\{\mathcal{X}^v\}_{v=1}^V$), which could be termed as a possibility for each category. Secondly, the uncertainty of each view ($\{u^v\}_{v=1}^V$) and the conflicts ($\{o^v\}_{v=1}^V$) between views are calculated based on these memberships. Thirdly, the weights $\{w^v\}_{v=1}^V$ of each view could be calculated and are used to aggregate memberships from all views, thereby realizing Dual-reliable Multi-view Fusion (DRF). Finally, the aggregated memberships are used to construct trusted classification results, where the decision uncertainty will increase if aggregated memberships are conflicting.

where $\mathbf{a}_i^v$ denotes the logits of a neural network. The corresponding category credibility $\mathbf{c}_i^v$ can also be derived by Equation (2).

To achieve discriminative learning, each sample should exhibit the highest possible category credibility for its matched category while maintaining the lowest possible category credibility for all other categories. Intuitively, this could be achieved by directly aligning the category credibility $\mathbf{c}_i^v$ with the corresponding one-hot labels $\mathbf{y}_i^v$, i.e., minimizing mean squared error ($\|\mathbf{c}_i^v - \mathbf{y}_i^v\|_2$) or cross-entropy loss ($-\mathbf{y}_i^v \cdot \log(\mathbf{c}_i^v) - (1 - \mathbf{y}_i^v) \cdot \log(1 - \mathbf{c}_i^v)$). However, both approaches risk over-optimizing the necessity ($\mathbf{e}_i^v$) for unmatched categories, resulting in the neural network converging to a local optimum. The reason is as follows: When $y_{ik}^v = 0$, $m_{ik}^v$ would tend to 0, and the necessity $e_{ik}^v = 1 - \max\{m_{il}^v \mid l \neq k\}$ would also approach 0, driving $m_{il}^v$ towards 1. This is problematic because $m_{il}^v$ should approach 0 when $y_{il}^v = 0$, rather than 1. To tackle this issue, we propose category credibility learning loss to optimize the category credibility, thereby guiding the model toward the correct optimization:

$$\mathcal{L}_{ccl} = \frac{1}{N_b} \sum_{i=1}^{N_b} -\mathbf{y}_i^v \cdot \log(\mathbf{r}_i^v) - (1 - \mathbf{y}_i^v) \cdot \log(1 - \mathbf{r}_i^v), \quad (4)$$

where $N_b$ is the batch size, $\mathbf{r}_i^v = \phi^{tr}(\mathbf{m}_i^v, \mathbf{y}_i^v) = [r_{i1}^v, r_{i2}^v, ..., r_{iK}^v] \in \mathbb{R}^K$ represents category credibility during training, and

$$r_{ik}^v = \begin{cases} \dfrac{m_{ik}^v + 1 - \max\{m_{il}^v \mid l \neq k\}}{2}, & \text{if } y_{ik}^v = 1, \\[3mm] \dfrac{m_{ik}^v + 1 - m_{il}^v}{2}, & \text{if } y_{ik}^v = 0, \ l = \arg\max_k \ y_{ik}^v, \end{cases} \quad (5)$$

where k = 1, 2, ..., K. From Equation (4), it can be seen

that this loss function ensure $m_{ik}^v$ approaches 1 for matched categories where $y_{ik}^v = 1$ and approaches 0 for unmatched category where $y_{ik}^v = 0$, by leveraging label information. To be specific, when $y_{ik}^v = 0$ and $y_{il}^v = 1$ (where $l = \arg\max_k y_{ik}^v$), we expect the membership of the matched category to be greater than that of any unmatched categories after training. For matched categories, the necessity during training should be calculated as $1 - \max\{m_{il}^v \mid l \neq k\}$, driving $\max\{m_{il}^v \mid l \neq k\}$ toward 0. For unmatched categories, the necessity during training should be calculated as $1 - m_{il}^v$, forcing $m_{il}^v$ to approach 1. Therefore, this approach effectively guides the necessity and category credibility optimization, avoiding over-optimization and ensuring correct model convergence.

### 3.2.3. CONFLICTIVE MULTI-VIEW FUSION

Environmental factors, such as sensor failure, adverse weather conditions, and data communication issues, often introduce noisy and unaligned views in multi-view data, i.e., create conflicting views (Xiao, 2019a; Zhang et al., 2024). Addressing these issues is essential for enhancing the precision and robustness of multi-view classification. Noisy views generally exhibit high uncertainty, complicating accurate decision-making and potentially leading to erroneous decisions. Therefore, the influence of noisy views should be minimized in multi-view fusion to prioritize the contributions of cleaner views. In contrast, unaligned views tend to generate highly conflicting but low-uncertainty decisions. Reducing their misleading impact on the final decision and limiting their influence in the fusion process is also crucial. Therefore, below, we first define the uncertainty and then the conflict, and finally use them to build a multi-view fusion strategy that can resist conflicting views.

**Uncertainty Inference**. Although the category credibility reflects the uncertainty of a single predicted category, it fails to capture the uncertainty of the entire prediction outcome. To overcome this limitation, we take into account the category credibility of all categories to calculate decision uncertainty. To be specific, inspired by Shannon's entropy, which measures uncertainty arising from information deficiency, we define uncertainty as follows:

**Definition 3.2.** Let $\mathbf{c}_i^v = [c_{i1}^v, c_{i2}^v, ..., c_{iK}^v]$ be the vector of category credibility of the $i$-th sample in $v$-th view, and $\forall c_{ik}^v \in [0, 1], \ k = 1, 2, ..., K$. Then, uncertainty is defined by

$$
\begin{aligned}
u_i^v &= \frac{\sum_{k=1}^{K} H(c_{ik}^v)}{K \cdot \ln 2} \\
&= \frac{\sum_{k=1}^{K} -c_{ik}^v \cdot ln(c_{ik}^v) - (1 - c_{ik}^v) \cdot ln(1 - c_{ik}^v)}{K \cdot \ln 2},
\end{aligned}
\tag{6}
$$

where $K$ is the number of categories and $H(c_{ik}^v)$ is the entropy of category credibility $c_{ik}^v$. This uncertainty lies within the range $[0, 1]$, where higher values denote greater uncertainty.

**Conflict Inference**. The above-defined uncertainty does not allow for the assessment of inconsistencies between views. To address this, below, we define conflict:

**Definition 3.3.** Let $\{\mathbf{m}_i^v\}_{v=1}^V$ represent the set of multi-view membership vectors for the $i$-th sample. The conflict of the $v$-th view relative to other views is defined as:

$$
o_i^v = \frac{1}{V-1} \sum_{j \neq v}^{V} \left( 1 - \frac{\mathbf{m}_i^v \cdot \mathbf{m}_i^j}{||\mathbf{m}_i^v|| \cdot ||\mathbf{m}_i^j||} \right).
\tag{7}
$$

Since the range of all elements in $\mathbf{m}_i^v$ is in the range [0,1], the conflict $o_i^v$ is in the range [0,1], where higher values denote greater conflict.

Views with relatively high conflict are considered unaligned with other views, whereas views with relatively low conflict are precisely the ones that should participate in the fusion. Next, we utilize view-specific uncertainty and the above conflict between views to propose a Dual-reliable Multi-view Fusion, which means both noisy views and unaligned views can be reliably fused.

**Dual-reliable Multi-view Fusion** (DRF). In multi-view decision-level fusion, we hope to fuse views with low uncertainty and low conflict with other views. Following Definition 3.2 and Definition 3.3, we could fuse the final memberships $\mathbf{m}_i^a$ from different views as follows:

$$
\begin{aligned}
w_i^v &= \frac{g\left((1 - u_i^v)(1 - o_i^v)\right)}{\sum_{v=1}^{V} g\left((1 - u_i^v)(1 - o_i^v)\right)}, \\
\mathbf{m}_i^a &= \sum_{v=1}^{V} w_i^v \cdot \mathbf{m}_i^v,
\end{aligned}
\tag{8}
$$

where $g(\cdot)$ is a monotonically increasing function. In this work, we use the exponential function $exp(\cdot)$ as $g(\cdot)$. Note that, during training, the uncertainty is calculated based on the category credibility during training, i.e., Equation (5). According to DRF, we could get the final memberships of each category and thus infer the overall uncertainty by Equation (6).

### 3.2.4. LOSS FUNCTION

To ensure that all views can simultaneously form reasonable decisions and thus improve the overall performance, we use a multi-task strategy with the following overall loss function:

$$
\mathcal{L}_{total} = \mathcal{L}_{ccl}(\mathbf{r}^a, \mathbf{y}) + \sum_{v=1}^{V} \mathcal{L}_{ccl}(\mathbf{r}^v, \mathbf{y}).
\tag{9}
$$

The pseudo-code of our FUML can be found in the Appendix B.3.

### 3.3. Discussion and Analyses

In this section, we analyze the advantages of FUML, especially the conflicting view fusion. The following propositions provide the theoretical analysis to support the conclusions. The proofs are shown in the Appendix A.

**Proposition 3.4.** *For the $i$-th multi-view instance, when a clean view 1 with uncertainty $u_i^1$ is fused with a conflicting view 2 caused by noise with uncertainty $u_i^2$, the fused uncertainty $u_i^a > u_i^1$.*

**Proposition 3.5.** *For the $i$-th multi-view instance, when a clean view 1 with uncertainty $u_i^1$ is fused with a conflicting view 2 caused by misalignment with uncertainty $u_i^2$, the fused uncertainty $u_i^a > u_i^1$ and $u_i^a > u_i^2$.*

Based on Proposition 3.4 and Proposition 3.5, during the multi-view fusion stage, our FUML achieves more accurate uncertainty estimation when fusing with a conflicting view.

**Intuitive explanation of the effectiveness of FUML.** Without loss of generality, we assume that view $\mathcal{X}^A$ is clean, view $\mathcal{X}^B$ is noisy due to unknown environmental factors or sensor failure, and view $\mathcal{X}^C$ is misaligned with other views for similar reasons. At this time, $\mathcal{X}^A$ aligns with the distribution of clean training data, whereas view $\mathcal{X}^B$ deviates significantly from it. Accordingly, we have $u^A \leqslant u^B$ and $o^A \leqslant o^B$, leading to $w^A \geqslant w^B$. Therefore, in our FUML framework, the multi-view decision will tend to rely more on the high-quality view $\mathcal{X}^A$ than on the noisy view $\mathcal{X}^B$. In addition, although the uncertainty of view $\mathcal{X}^C$ is not as high as that of view $\mathcal{X}^B$, its relative view $\mathcal{X}^C$ has a higher conflict with other views. Accordingly, we have $u^A \approx u^C$ and $o^A \leqslant o^C$ thus $w^A \geqslant w^C$. Therefore, for our method, the multi-view decision will tend to rely more on the low-conflict view $\mathcal{X}^A$ than the view $\mathcal{X}^C$. As for the

weight between views $\mathcal{X}^B$ and view $\mathcal{X}^C$, it is related to the comprehensive consideration of its uncertainty and conflict. By dynamically determining the fusion weights of each view, FUML effectively mitigates the influence of noisy and misaligned views, i.e., conflicting views, embracing robust classification for conflicting multi-view instances.

# 4. Experiments

## 4.1. Experimental Setup

We briefly present the experimental setup here, including the experimental datasets and comparison methods. Please refer to Appendix B for more detailed setup. The code of our FUML is available here[1].

**Datasets.** To validate the effectiveness of the proposed FUML, we conduct experiments on eight public datasets: **Handwritten** (HW)[2], **MSRC-V1** (MSRC) (Winn & Jojic, 2005), **NUS-WIDE-OBJ** (NUSOBJ)[3], **Fashion-MV** (Fashion) (Wang et al., 2023), **Scene15** (Scene)[4], **LandUse** (Yang & Newsam, 2010), **Leaves100** (Leaves)[5], and **PIE**[6]. The training set and the test set are split in a ratio of 8:2. The detailed information is shown in Table 1.

To create a test set with conflicting instances, following the methodology outlined in (Xu et al., 2024a), we apply two transformations: **(1)** We add Gaussian noise with different standard deviations to some test instances. **(2)** We alter the information in a random view for a subset of instances, making the view's label inconsistent with the true label.

*Table 1.* A summary of datasets used for evaluation.

| DATASET | SIZE | CATEGORIES | DIMENSIONALITY |
|---|---|---|---|
| HW | 2000 | 10 | 240; 76; 216; 47; 64; 6 |
| MSRC | 210 | 7 | 1302; 48; 512; 100; 256; 210 |
| NUSOBJ | 30000 | 31 | 65; 226; 145; 74; 129 |
| FASHION | 10000 | 10 | 784; 784; 784 |
| SCENE | 4485 | 15 | 20; 59; 40 |
| LANDUSE | 2100 | 21 | 20; 59; 40 |
| LEAVES | 1600 | 100 | 64; 64; 64 |
| PIE | 680 | 68 | 484; 256; 279 |

**Evaluation metrics.** Owing to the inherent randomness, we report the mean accuracy and standard deviation across 10 different random seeds. Additionally, the improvements over the best-performing baseline are also reported.

**Compared methods.** For a comprehensive comparison, we adopted the following baselines: **(1)** The untrusted

[1]https://github.com/siyuancncd/FUML

[2]https://archive.ics.uci.edu/ml/datasets/Multiple+Features

[3]https://lms.comp.nus.edu.sg/wp-content/uploads/2019/

[4]https://doi.org/10.6084/m9.figshare.7007177.v1

[5]https://archive.ics.uci.edu/dataset/241/one+hundred+plant+species+leaves+data+set

[6]http://www.cs.cmu.edu/afs/cs/project/PIE/MultiPie/MultiPie/Home.html

baselines, i.e., can't provide decision uncertainty, include: **DFTMC** (Han et al., 2022a), **DCP(CV&CG)** (Lin et al., 2022), **QMF** (Zhang et al., 2023), and **PDF** (Cao et al., 2024). **(2)** The trusted baselines include: **DUA-Nets** (Geng et al., 2021), **TMC** (Han et al., 2020), **ETMC** (Han et al., 2022b), **TMDL-OA** (Liu et al., 2022), **UIMC** (Xie et al., 2023), **ECML** (Xu et al., 2024a), **TMNR** (Xu et al., 2024b), and **CCML** (Liu et al., 2024).

## 4.2. Comparison with State-of-the-Art Methods

To evaluate the performance of our FUML, we apply multi-view classification on eight datasets over 10 different random seeds. The experimental results of the normal and conflicting test sets are shown in Table 2 and Table 3, respectively. Note that DFTMC does not converge on the NUSOBJ and Fashion datasets, so we can't report its results and mark with '-'. The following key observations can be made from these results: **(1)** On the normal test sets, FUML outperforms all baselines on all datasets. For instance, on the Scene, Leaves, and PIE datasets, FUML achieves an accuracy improvement of 1.62%, 1.59%, and 1.47% compared to the second-best baselines. **(2)** When the performance is compared on the conflicting test sets, all methods exhibit a noticeable drop in accuracy. However, FUML consistently achieves superior performance, with particularly larger improvements on the Scene, LandUse, and Leaves datasets (4.83%, 7.31%, and 14.60%, respectively). This could be attributed to the proposed dual-reliable multi-view fusion, which improves the robustness to conflicting multi-view instances by reducing the weights of noisy and unaligned views during fusion, as verified by the ablation studies in Section 4.5. More comprehensive conflicting multi-view classification results and analysis can be found in Appendix C.1.

## 4.3. Uncertainty Effectiveness Analysis

To validate the effectiveness of our FUML in estimating uncertainty for conflicting multi-view instances, we compare it with two typical EDL-based TMVC methods, i.e., ETMC and ECML, using the uncertainty density map. The results, shown in Figure 3, reveal the following observations: **(1)** Compared to the normal test set, the uncertainty of ETMC and ECML barely changed with the addition of conflicting views, and even decreased on the LandUse dataset. However, as can be seen from Table 2 and Table 3, for ETMC and ECML, the addition of conflicting views greatly reduces the classification accuracy. Therefore, their uncertainty estimates for conflicting multi-view instances are inaccurate. **(2)** In contrast, the uncertainty estimated by FUML is notably higher for conflicting test sets than for normal ones, demonstrating the accuracy of FUML in estimating uncertainty since it can facilitate discrimination between normal and conflicting instances. The corresponding quantitative

*Table 2.* Accuracy (%) performance on normal test sets. The best and the second-best results are highlighted in **boldface** and underlined respectively. The means and standard deviations over ten runs are reported. The methods marked with '★' are trusted.

| METHODS. | HW | MSRC | NUSOBJ | FASHION | SCENE | LANDUSE | LEAVES | PIE |
|---|---|---|---|---|---|---|---|---|
| DFTMC | 98.75±0.39 | 96.90±2.14 | - | - | 63.10±3.60 | 34.95±1.69 | 69.92±2.54 | 91.40±3.50 |
| DCP-CV | 98.75±0.59 | 92.86±2.61 | 32.19±9.48 | 97.96±0.16 | 76.70±2.15 | 71.71±2.09 | 95.62±1.38 | 86.32±4.87 |
| DCP-CG | 99.00±0.47 | 95.24±3.69 | 43.65±1.10 | 98.11±0.23 | 77.79±1.73 | 75.74±0.98 | 98.19±0.46 | 90.59±1.99 |
| QMF | 98.72±0.48 | 97.86±1.28 | 45.41±0.43 | 98.93±0.32 | 68.58±1.49 | 47.86±2.55 | 95.69±1.25 | 92.06±1.64 |
| PDF | 98.40±0.37 | 97.14±1.78 | 46.78±0.33 | 98.95±0.19 | 70.25±1.21 | 45.17±2.66 | 98.03±0.71 | 92.57±1.66 |
| DUA-NETS★ | 98.10±0.32 | 84.67±3.03 | 27.75±0.00 | 91.08±0.17 | 65.01±1.55 | 45.24±1.85 | 90.31±1.25 | 90.56±0.47 |
| TMC★ | 98.51±0.15 | 91.70±2.70 | 38.77±0.81 | 95.40±0.40 | 67.71±0.30 | 31.69±3.93 | 86.81±2.20 | 91.85±0.23 |
| ETMC★ | 98.75±0.00 | 92.86±3.01 | 44.23±0.76 | 96.21±0.36 | 71.61±0.28 | 43.52±3.19 | 91.44±2.39 | 93.75±1.08 |
| TMDL-OA★ | 98.55±0.45 | 95.00±1.67 | 27.88±0.67 | 86.52±0.04 | 75.57±0.02 | 25.02±2.10 | 75.28±3.57 | 92.33±0.36 |
| UIMC★ | 98.25±0.00 | 98.81±1.19 | 43.42±0.12 | 98.13±0.13 | 77.70±0.00 | 57.95±0.61 | 95.31±0.71 | 91.69±2.16 |
| ECML★ | 98.72±0.39 | 94.05±1.60 | 42.62±0.42 | 97.93±0.35 | 76.19±0.12 | 60.10±2.01 | 92.53±1.94 | 94.71±0.02 |
| TMNR★ | 97.20±0.63 | 94.05±3.24 | 34.52±0.85 | 94.10±0.50 | 68.10±1.15 | 27.38±1.88 | 90.13±1.53 | 89.53±1.89 |
| CCML★ | 97.60±0.62 | 96.90±2.39 | 41.43±0.71 | 95.16±0.41 | 73.87±1.83 | 60.86±1.93 | 97.72±0.92 | 93.97±1.67 |
| FUML★ | **99.20±0.36** | **99.76±0.75** | **48.23±0.42** | **98.96±0.25** | **79.41±1.34** | **76.71±0.46** | **99.78±0.27** | **96.18±1.24** |
| IMPROVE | Δ 0.20 | Δ 0.95 | Δ 1.45 | Δ 0.01 | Δ 1.62 | Δ 0.97 | Δ 1.59 | Δ 1.47 |

*Table 3.* Accuracy (%) performance on conflicting test sets. The best and the second-best results are highlighted in **boldface** and underlined respectively. The means and standard deviations over ten runs are reported. The methods marked with '★' are trusted.

| METHODS | HW | MSRC | NUSOBJ | FASHION | SCENE | LANDUSE | LEAVES | PIE |
|---|---|---|---|---|---|---|---|---|
| DFTMC | 53.65±20.07 | 60.24±23.45 | - | - | 36.01±2.78 | 7.88±0.94 | 1.10±0.12 | 3.97±0.82 |
| DCP-CV | 98.20±0.56 | 84.76±7.00 | 28.10±7.80 | 92.72±2.41 | 66.22±2.12 | 59.98±1.93 | 76.94±1.36 | 67.06±2.15 |
| DCP-CG | 98.70±0.64 | 90.00±1.78 | 38.61±1.29 | 90.38±2.17 | 66.44±0.32 | 61.83±2.48 | 79.06±1.22 | 69.56±3.77 |
| QMF | 97.52±0.86 | 95.95±1.52 | 42.72±0.67 | 92.69±0.78 | 59.53±1.63 | 40.17±2.67 | 77.47±1.46 | 82.50±2.81 |
| PDF | 94.35±1.21 | 94.52±3.02 | 43.57±0.36 | 90.73±0.53 | 58.75±1.03 | 39.40±1.94 | 76.34±1.26 | 74.93±2.76 |
| DUA-NETS★ | 87.16±0.34 | 78.57±4.45 | 25.64±0.25 | 83.03±0.18 | 26.18±1.31 | 37.22±0.56 | 65.62±2.19 | 56.45±1.75 |
| TMC★ | 92.76±0.15 | 86.20±4.90 | 36.00±0.78 | 84.76±0.78 | 42.27±1.61 | 19.67±1.88 | 70.25±2.55 | 61.65±1.03 |
| ETMC★ | 93.85±1.26 | 87.14±4.54 | 40.45±0.81 | 86.48±1.05 | 56.90±1.70 | 36.05±2.50 | 74.19±1.74 | 73.82±4.77 |
| TMDL-OA★ | 92.45±0.05 | 84.52±2.20 | 27.02±0.75 | 74.55±0.07 | 48.42±1.02 | 21.71±1.83 | 62.28±3.70 | 68.16±0.34 |
| UIMC★ | 97.72±0.18 | 96.43±1.19 | 41.72±0.31 | 89.71±0.25 | 67.88±0.48 | 50.43±0.46 | 79.84±0.92 | 70.66±2.04 |
| ECML★ | 94.52±0.79 | 90.00±2.78 | 39.89±0.59 | 84.02±0.51 | 56.97±0.52 | 50.31±1.81 | 74.88±1.89 | 84.00±0.14 |
| TMNR★ | 92.78±1.01 | 90.71±4.19 | 30.88±0.58 | 85.76±0.81 | 60.00±1.43 | 23.95±1.92 | 74.09±1.99 | 80.59±3.26 |
| CCML★ | 93.22±1.09 | 94.29±2.18 | 37.38±0.65 | 83.84±1.01 | 62.08±1.34 | 52.48±2.74 | 78.87±2.31 | 83.24±2.79 |
| FUML★ | **98.78±0.36** | **98.81±1.60** | **47.08±0.32** | **96.68±0.32** | **72.71±1.75** | **69.14±2.43** | **94.44±1.18** | **88.01±2.53** |
| IMPROVE | Δ 0.08 | Δ 2.38 | Δ 3.51 | Δ 3.96 | Δ 4.83 | Δ 7.31 | Δ 14.60 | Δ 4.01 |

(a) ETMC (Fashion)  (b) ECML (Fashion)  (c) FUML (Fashion)  (d) ETMC (LandUse)  (e) ECML (LandUse)  (f) FUML (LandUse)

*Figure 3.* Density of uncertainty on the normal and conflicting test sets of the Fashion and LandUse datasets.

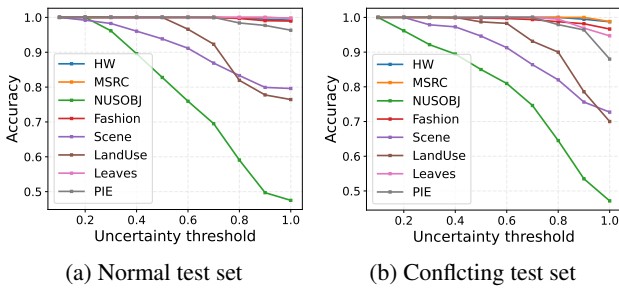

(a) Normal test set  (b) Conflcting test set

*Figure 4.* Accuracy with uncertainty thresholding.

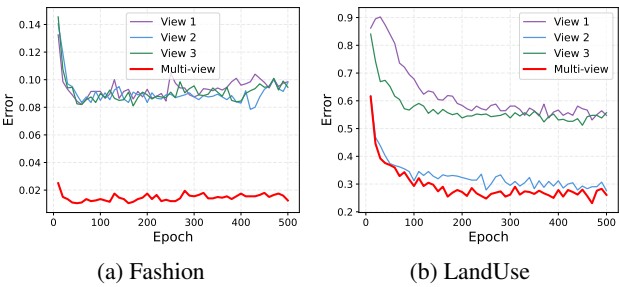

(a) Fashion  (b) LandUse

*Figure 5.* Prediction error with different epochs.

results can be found in Appendix C.2.

Additionally, to observe the trend of classification accuracy of FUML as the uncertainty threshold varies, we plot Figure 4. It illustrates that FUML achieves significantly more accurate predictions as the prediction uncertainty decreases for both normal and conflicting test sets on all eight datasets. This demonstrates that our model's output, i.e., classification results and the corresponding uncertainty, supports making trusted decisions.

### 4.4. Multi-view Fusion Effectiveness Evaluation

To evaluate the effectiveness of our FUML for multi-view fusion, we compare the prediction error of multi-view learning results (depicted as a red line, labeled "Multi-view") with the prediction error of each single-view learning result on the Fashion and LandUse datasets. As shown in Figure 5, the prediction error of the multi-view is consistently lower than that of any single-view in the proposed method, demonstrating that it effectively reduces prediction error by integrating multiple views to achieve more accurate results. Results for the other six datasets are provided in Appendix C.3.

### 4.5. Ablation Study

To demonstrate the effectiveness of each component of our FUML, we perform an ablation study on the conflicting test set of the Fashion and LandUse datasets with mean accuracy (Acc.), mean precision (Prec.), and mean F-score

*Table 4.* Ablation study on the conflicting test sets of the Fashion and LandUse datasets with all metrics in percentages (%).

| METHOD | | | FASHION | | | LANDUSE | | |
|---|---|---|---|---|---|---|---|---|
| $\mathcal{L}_{ccl}^a$ | $\mathcal{L}_{ccl}^v$ | RULE | ACC. ↑ | PREC. ↑ | F-SCORE ↑ | ACC. ↑ | PREC. ↑ | F-SCORE ↑ |
| √ | × | DRF | 96.33 | 96.31 | 96.30 | 68.00 | 68.72 | 67.53 |
| × | √ | DRF | 96.32 | 96.32 | 96.30 | 67.76 | 68.41 | 67.33 |
| √ | × | CONCAT | 88.45 | 88.28 | 88.75 | 59.69 | 60.54 | 58.54 |
| √ | √ | AVG | 96.15 | 96.13 | 96.13 | 67.71 | 68.22 | 67.42 |
| √ | √ | DRF | **96.68** | **96.68** | **96.68** | **69.14** | **70.21** | **69.19** |

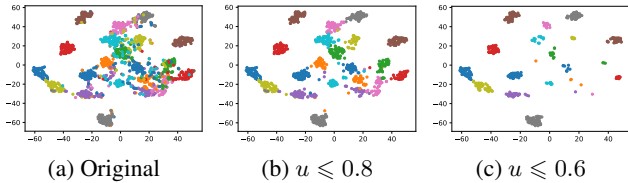

(a) Original  (b) $u \leqslant 0.8$  (c) $u \leqslant 0.6$

*Figure 6.* Visualization of aggregated memberships on the training set of LandUse dataset by $t$-SNE (Van der Maaten & Hinton, 2008). Samples belonging to the same category are rendered with the same color. (a) Display the results of fused memberships. (b)-(c) demonstrate the results of fused memberships after the samples with uncertainty greater than 0.8 and 0.6 are removed, respectively.

over 10 seeds. To be specific, we calculate metrics for each label and find their unweighted mean for precision and F-score. In addition, for simplicity, in this section, we represent $\mathcal{L}_{fcl}(\mathbf{r}^a, \mathbf{y})$ as $\mathcal{L}_{ccl}^a$ and represent $\sum_{v=1}^{V} \mathcal{L}_{fcl}(\mathbf{r}^v, \mathbf{y})$ as $\mathcal{L}_{ccl}^v$. The results, presented in Table 4, reveal: **(1)** After removing $\mathcal{L}_{ccl}^a$ or $\mathcal{L}_{ccl}^v$, all indicators decline to varying degrees, which shows that all components in the total loss function are indispensable. **(2)** When performing feature-level fusion, i.e., concatenating (Concat) all features and only using one DNN for prediction, performance declines significantly. **(3)** Compared to the arithmetic mean (Avg), our DRF fusion demonstrates superior performance. In summary, all components of FUML are indispensable. Additional ablation experimental results on the normal test set can be found in Appendix C.4.

### 4.6. *t*-SNE Visualization and Analysis.

We employ the $t$-SNE approach to embed the fused memberships of the training set from the LandUse dataset into a two-dimensional visualization plane, as shown in Figure 6. The results demonstrate: **(1)** Distinct categories occupy different Spaces, and are well distinguished, indicating that FUML learns discriminative information. **(2)** After filtering out samples with uncertainty greater than 0.8 and 0.6, the category boundaries gradually become more distinct. These improvements are attributed to the effective uncertainty estimation capability of FUML.

# 5. Conclusion

This paper presents the Deep Fuzzy Multi-view Learning method (FUML), a novel multi-view classification framework designed to accurately classify conflicting multi-view instances and precisely estimate their intrinsic uncertainty. Based on the Fuzzy Set Theory, our FUML models the outputs of classification neural networks as a set of fuzzy memberships and quantifies category credibility by incorporating both possibility and necessity measures. To optimize category credibility, we propose a category credibility learning loss. In addition, we propose a Dual-reliable Fusion (DRF) strategy, which assigns weights based on view-specific uncertainty and inter-view conflict that effectively mitigates the influence of conflicting views. Extensive experiments on eight public datasets demonstrate FUML's superiority over 13 state-of-the-art methods in terms of accuracy, robustness, and reliability, particularly in challenging scenarios with conflicting views.

# Acknowledgements

This work was supported in part by the National Key R&D Program of China 2024YFB4710604; in part by NSFC under Grant 62472295, 62176171, 62372315, and U24B20174; in part by the Fundamental Research Funds for the Central Universities under Grant CJ202303, and CJ202403; in part by Sichuan Science and Technology Planning Project under Grant 2024ZDZX0004, 2024NS-FTD0049, 2024ZHCG0005, and 24NSFTD0130; in part by TCL science and technology innovation fund; in part by System of Systems and Artificial Intelligence Laboratory pioneer fund grant; and in part by the Chengdu Science and Technology Project under Grant 2023-XT00-00004-GX.

# Impact Statement

This paper presents work to advance the field of multi-view learning in machine learning. Our goal is to construct a trusted and reliable multi-view classification method to boost the accuracy and credibility of joint decisions in multi-view systems, lowering the potential classification errors and inaccurate uncertainty estimates of prediction. However, due to the data bias in open environments, there is a possibility of inevitable error when applying our method in real-world applications.

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

# APPENDIX

This document provides mathematical proofs, additional experimental details, additional experimental results, and limitations to support the paper:

## A. Proofs

### A.1. Proof of Proposition 3.4

*Proof.* Without loss of generality, for the $i$-th multi-view instance, let $\mathbf{m}_i^1 = \left[m_{i1}^1, m_{i2}^1, ..., m_{iK}^1\right]$ denote the sorted memberships of a clean view $\mathcal{X}_i^1$, i.e., $m_{i1}^1 \geqslant m_{i2}^1 \geqslant, ..., m_{iK}^1$, and $\mathbf{m}_i^2 = \left[m_{i1}^2, m_{i2}^2, ..., m_{iK}^2\right]$ denote sorted memberships of another noisy view $\mathcal{X}_i^2$. We assume that $m_{i1}^1 - m_{ik}^1 \geqslant m_{i1}^2 - m_{ik}^2, k = 2, 3, ..., K$. Therefore, we have

$$
\begin{aligned}
c_{i1}^1 &= \frac{m_{i1}^1 + 1 - m_{i2}^1}{2}, \\
c_{ik}^1 &= \frac{m_{ik}^1 + 1 - m_{i1}^1}{2}, \ k = 2, 3, ..., K, \\
u_i^1 &= \frac{H(\frac{m_{i1}^1 + 1 - m_{i2}^1}{2}) + \sum_{k=2}^K H(\frac{m_{ik}^1 + 1 - m_{i1}^1}{2})}{K \cdot \ln 2},
\end{aligned}
\tag{10}
$$

where $H(t) = -t \cdot ln(t) - (1 - t) \cdot ln(1 - t)$. Because $t = 0.5$ is the symmetry axis of $H(t)$, we have

$$
u_i^1 = \frac{H(\frac{m_{i1}^1 - m_{i2}^1 + 1}{2}) + \sum_{k=2}^K H(\frac{m_{i1}^1 - m_{ik}^1 + 1}{2})}{K \cdot \ln 2}
\tag{11}
$$

After fusing view $\mathcal{X}_i^1$ and $\mathcal{X}_i^2$, we have

$$
\mathbf{m}_{ik}^a = w_i^1 \cdot \mathbf{m}_{ik}^1 + w_i^2 \cdot \mathbf{m}_{ik}^2, \ k = 2, 3, ..., K,
\tag{12}
$$

where $\mathbf{m}_i^a$ denotes the fused memberships, and $w_i^1 > 0$ and $w_i^2 > 0$ represent the weights of view $\mathcal{X}_i^1$ and $\mathcal{X}_i^2$, respectively. Therefore, we have

$$
\begin{aligned}
m_{i1}^a - m_{i2}^a &= w_i^1 \cdot (m_{i1}^1 - m_{i2}^1) + w_i^2 \cdot (m_{i1}^2 - m_{i2}^2) < m_{i1}^1 - m_{i2}^1, \\
m_{i1}^a - m_{ik}^a &= w_i^1 \cdot (m_{i1}^1 - m_{ik}^1) + w_i^2 \cdot (m_{i1}^2 - m_{ik}^2) < m_{i1}^1 - m_{ik}^1, \ k = 2, 3, ..., K,
\end{aligned}
\tag{13}
$$

and

$$H(\frac{m_{i1}^a - m_{i2}^a + 1}{2}) > H(\frac{m_{i1}^1 - m_{i2}^1 + 1}{2}),$$
$$H(\frac{m_{i1}^a - m_{ik}^a + 1}{2}) > H(\frac{m_{i1}^1 - m_{ik}^1 + 1}{2}), \ k = 2, 3, ..., K. \tag{14}$$

Therefore, we have

$$u_i^a = \frac{H(\frac{m_{i1}^a - m_{i2}^a + 1}{2}) + \sum_{k=2}^{K} H(\frac{m_{i1}^a - m_{ik}^a + 1}{2})}{K \cdot \ln 2} > \frac{H(\frac{m_{i1}^1 - m_{i2}^1 + 1}{2}) + \sum_{k=2}^{K} H(\frac{m_{i1}^1 - m_{ik}^1 + 1}{2})}{K \cdot \ln 2} = u_i^1. \tag{15}$$

$\square$

## A.2. Proof of Proposition 3.5

*Proof.* Without loss of generality, for the $i$-th multi-view instance, let $\mathbf{m}_i^1 = [m_{i1}^1, m_{i2}^1, ..., m_{iK}^1]$ denote the memberships of a clean view $\mathcal{X}_i^1$, and $\mathbf{m}_i^2 = [m_{i1}^2, m_{i2}^2, ..., m_{iK}^2]$ denote the memberships of another clean but misaligned view $\mathcal{X}_i^2$. We assume that the view $\mathcal{X}_i^1$ is of the $p$-th category and the view $\mathcal{X}_i^2$ is of the $q$-th category. Therefore, we can assume that

$$m_{ip}^1 \gg m_{ih}^1, \ \forall h \neq p,$$
$$m_{iq}^2 \gg m_{id}^2, \ \forall d \neq q,$$
$$m_{ip}^1 \approx m_{iq}^2. \tag{16}$$

After that, for view $\mathcal{X}_i^1$ we have

$$c_{ip}^1 = \frac{m_{ip}^1 + 1 - \max\{m_{ih}^1 | h \neq p\}}{2},$$
$$c_{ih}^1 = \frac{m_{ih}^1 + 1 - m_{ip}^1}{2}. \tag{17}$$

Then,

$$u_i^1 = \frac{H(c_{ip}^1) + \sum_{h \neq p} H(c_{ih}^1)}{K \cdot \ln 2} = \frac{H(\frac{m_{ip}^1 + 1 - \max\{m_{ih}^1 | h \neq p\}}{2}) + \sum_{k \neq p} H(\frac{m_{ih}^1 + 1 - m_{ip}^1}{2})}{K \cdot \ln 2}, \tag{18}$$

where $H(t) = -t \cdot ln(t) - (1 - t) \cdot ln(1 - t)$. Because $t = 0.5$ is the symmetry axis of $H(t)$, we have

$$u_i^1 = \frac{H(\frac{m_{ip}^1 + 1 - \max\{m_{ih}^1 | h \neq p\}}{2}) + \sum_{h \neq p} H(\frac{m_{ip}^1 + 1 - m_{ih}^1}{2})}{K \cdot \ln 2}, \tag{19}$$

Meanwhile, for view $\mathcal{X}_i^2$, we have

$$u_i^2 = \frac{H(\frac{m_{iq}^2 + 1 - \max\{m_{ih}^2 | d \neq q\}}{2}) + \sum_{d \neq q} H(\frac{m_{iq}^2 + 1 - m_{id}^2}{2})}{K \cdot \ln 2}. \tag{20}$$

After fusing view $\mathcal{X}_i^1$ and $\mathcal{X}_i^2$, because $m_{ip}^1 \approx m_{iq}^2 \gg m_{iq}^1, m_{ip}^2, m_{ik}^1, m_{ik}^2, k \neq p$ and $k \neq q$, we have

$$u_i^a = \frac{H(\frac{m_{ip}^a + 1 - m_{iq}^a}{2}) + H(\frac{m_{iq}^a + 1 - m_{ip}^a}{2}) + \sum_{k \neq p, k \neq q} H(\frac{m_{ik}^a + 1 - m_{ip}^a}{2})}{K \cdot \ln 2},$$
$$\approx \frac{2 \cdot \ln 2 + \sum_{k \neq p, k \neq q} H(\frac{m_{ip}^a + 1 - m_{ik}^a}{2})}{K \cdot \ln 2} \tag{21}$$

Next, we prove that $u_i^a > u_i^1$. Specifically, we have

$$m_{ip}^a - m_{ik}^a = w_i^1 \cdot (m_{ip}^1 - m_{ik}^1) + w_i^2 \cdot (m_{ip}^2 - m_{ik}^2), \tag{22}$$

where $w_i^1 \in (0,1)$ and $w_i^2 \in (0,1)$ represent the weights of view $\mathcal{X}_i^1$ and $\mathcal{X}_i^2$, respectively. In addition, we have $w_i^1 + w_i^2 = 1$. Because $m_{ip}^1 > m_{iq}^2 \gg m_{iq}^1, m_{ip}^2, m_{ik}^1, m_{ik}^2, k \neq p$ and $k \neq q$, we can deduce that

$$m_{ip}^2 - m_{ik}^2 < m_{ip}^a - m_{ik}^a < m_{ip}^1 - m_{ik}^1, \forall k \neq p. \tag{23}$$

Therefore,

$$H(\frac{m_{ip}^1 + 1 - m_{ik}^1}{2}) < H(\frac{m_{ip}^a + 1 - m_{ik}^a}{2}). \tag{24}$$

Then, we can deduce that

$$H(\frac{m_{ip}^1 + 1 - \max\{m_{ik}^1 | k \neq p\}}{2}) \ll H(\frac{1}{2}) = \ln 2. \tag{25}$$

Combine Equations (19), (21), (24) and (25), we can deduce that

$$u_i^1 = \frac{H(\frac{m_{ip}^1 + 1 - \max\{m_{ih}^1 | h \neq p\}}{2}) + \sum_{h \neq p} H(\frac{m_{ip}^1 + 1 - m_{ih}^1}{2})}{K \cdot \ln 2} < \frac{2 \cdot \ln 2 + \sum_{k \neq p, k \neq q} H(\frac{m_{ip}^a + 1 - m_{ik}^a}{2})}{K \cdot \ln 2} = u_i^a. \tag{26}$$

Similarly, $u^a > u^2$ can also be easily proved.

$\square$

## B. Experimental Details

### B.1. Datasets Details

The multi-view data used in this paper include:

- **HandWritten** (HW) [7] comprises 2000 instances of handwritten numerals ranging from '0' to '9', with 200 patterns per class, represented using six feature sets.

- **MSRC-V1** (MSRC) (Winn & Jojic, 2005) contains 210 images. Each image includes 7 classes. Following (Nie et al., 2017), we extract five features, including CM, HOG, GIST, CENTRIST feature, and LBP.

- **NUS-WIDE-OBJECT**[8] (NUSOBJ) consists of 30,000 images of 31 classes. Each instance is described as 5 views, including Color Histogram, block-wise Color Moments, Color Correlogram, Edge Direction Histogram, and Wavelet Texture.

- **Fashion-MV** (Fashion) (Wang et al., 2023) is an image dataset that contains 10 categories with a total of 30,000 fashion products. It has three views, each consisting of 10,000 grayscale images sampled from the same category.

- **Scene15** (Scene)[9] includes 4485 images from 15 indoor and outdoor scene categories, with features extracted using GIST, PHOG, and LBP.

- **LandUse** (Yang & Newsam, 2010) contains 2100 satellite images with 3 views and 21 categories.

- **Leaves100** (Leaves)[10] consists of 1600 leaf samples from 100 plant species. We extracted shape descriptors, fine-scale edges, and texture histograms as 3 views.

- **PIE**[11] contains 680 instances belonging to 68 classes, with intensity, LBP, and Gabor as 3 views.

---

[7]https://archive.ics.uci.edu/ml/datasets/Multiple+Features

[8]https://lms.comp.nus.edu.sg/wp-content/uploads/2019/

[9]https://doi.org/10.6084/m9.figshare.7007177.v1

[10]https://archive.ics.uci.edu/dataset/241/one+hundred+plant+species+leaves+data+set

[11]http://www.cs.cmu.edu/afs/cs/project/PIE/MultiPie/MultiPie/Home.html

## B.2. Baselines Details

We compare the proposed FUML with the following baselines:

**(1)** The untrusted baselines include:

- **DFTMC** (Dynamical Fusion for Trustworthy Multimodal Classification) (Han et al., 2022a) captures both feature and modality informativeness, proposing a dynamical fusion network for trustworthy multimodal classification.

- **DCP** (Dual Contrastive Prediction) (Lin et al., 2022) Provides an information-theoretic framework integrating consistency learning and data recovery, imputing missing views by minimizing conditional entropy through dual prediction.

- **QMF** (Quality-aware Multimodal Fusion) (Zhang et al., 2023) improves classification accuracy and model robustness by a provably robust multimodal fusion method.

- **PDF** (Predictive Dynamic Fusion) (Cao et al., 2024) reveals the multimodal fusion from a generalization perspective, improving reliability and stability.

**(2)** The trusted baselines include:

- **DUA-Nets** (Dynamic Uncertainty-Aware Networks) (Geng et al., 2021) utilizes reversal networks to integrate intrinsic information from different views into a unified representation.

- **TMC** (Trusted Multi-view Classification) (Han et al., 2020) pioneers addressing the uncertainty estimation problem in multi-view classification and producing trusted classification results.

- **ETMC** (Enhanced Trusted Multi-view Classification) (Han et al., 2022b) extends TMC by incorporating a pseudo view, enabling comprehensive interaction among different views.

- **TMDL-OA** (Trusted Multi-View Deep Learning with Opinion Aggregation) (Liu et al., 2022) proposes a consistency measure loss to achieve trustworthy learning results.

- **UIMC** (Uncertainty-induced Incomplete Multi-View Data Classification) (Xie et al., 2023) uses uncertainty-based imputation and evidence-based fusion for reliable classification of incomplete multi-view data.

- **ECML** (Evidential Conflictive Multiview Learning) (Xu et al., 2024a) is the SOTA method for conflict multi-view classification, which proposes a new opinion aggregation strategy.

- **TMNR** (Trusted Multi-view Learning with Label Noise) (Xu et al., 2024b) is a reliable multi-view learning model under the guidance of noisy labels.

- **CCML** (Consistent and Complementary-aware trusted Multi-view Learning) (Liu et al., 2024) solves the problem of data semantic fuzziness in multi-view learning by dynamically decoupling consistency and complementary evidence, thus improving the accuracy and reliability of classification.

## B.3. Implementation Details

All experiments are implemented in PyTorch and are carried out on NVIDIA Tesla V100S. During the training phase, our FUML uses Adam (Kingma & Ba, 2015) with $\beta_1 = 0.9$, $\beta_2 = 0.999$, a weight decay of 0.0001, and a maximum of 500 epochs. The $p$ in Equation (3) is set to 3. For the NUSOBJ and Fashion datasets, the learning rate is set to 0.0002 and the batch size to 400, while for the remaining six datasets, the learning rate is set to 0.001 and the batch size to 100. The pseudo-code of FUML is shown in Algorithm 1. In addition, for a fair comparison, we replace the backbone networks of QMF (Zhang et al., 2023) and PDF (Cao et al., 2024) with the same fully connected layer as FUML while preserving their core models and loss functions. For other baselines, we follow the settings in their source code.

---

**Algorithm 1** FUML algorithm

---

/*Training*/
**Input:** the multi-view training data $\{\{\mathbf{x}_n^v\}_{v=1}^V, \mathbf{y}_n\}_{n=1}^N$, batch size $N_b$, maximal epoch number $N_e$, learning rate $\eta$, and the multi-view model $\{f^v(\cdot, \theta^v)\}_{v=1}^V$.
**Output:** optimized network parameters $\{\theta^v\}_{v=1}^V$.
**for** $1, 2, \cdots, N_e$ **do**
    Randomly select $N_b$ samples from every view to construct a multi-view mini-batch.
    Calculate the output $\{\{\mathbf{a}_j^v\}_{j=1}^{N_b}\}_{v=1}^V$ for all samples of the mini-batch by using their corresponding model $\{f^v(\cdot, \theta^v)\}_{v=1}^V$.
    Calculate view-specific memberships $\{\mathbf{m}^v\}_{v=1}^V$ by Equation (3).
    Calculate credibility degrees during training $\{\mathbf{r}^v\}_{v=1}^V$ by Equation (5).
    Calculate category credibility during training by Equation (5) and aggregate memberships by Equation (8).
    Compute $\mathcal{L}_{total}$ according to Equation (9) on minibatch.
    Update FUML parameters $\{\theta^i\}_{i=1}^V$ using gradient descent algorithm with learning rate $\eta$.
**end for**
/*Testing*/
Calculate the view-specific memberships by the trained model.
Calculate category credibility by Equation (2) and fuse memberships by Equation (8).

---

## C. Additional Experiments

### C.1. Additional Conflicting Multi-view Classification Results and Analysis

To further assess the performance of our FUML in conflicting multi-view classification, we compare it against the three best-performing untrusted multi-view classification methods, i.e., DCP-CG, QMF, and PDF, as well as the three best-performing trusted multi-view classification methods, i.e., UIMC, ECML, and CCML. Each method is evaluated over 10 runs, and the mean values along with standard deviations are reported.

Table 9 presents the experimental results where Gaussian noise with a mean of 0 and variances of 1, 5, and 10 is randomly added to half of the views in half of the test sets. Similarly, Table 10 illustrates the results of adding Gaussian noise with a mean of 0 and a variance of 1 to half of the views in randomly selected 10%, 20%, 30%, 40%, and 50% of the test set. Finally, Table 11 displays the experimental results of introducing unaligned views to half of the views in randomly selected 10%, 20%, 30%, 40%, and 50% of the test sets. These results indicate that our FUML surpasses nearly all comparison methods across various conflict settings, highlighting its effectiveness in conflicting multi-view classification.

### C.2. Additional Uncertainty Effectiveness Analysis

In Section 4.4 and Figure 3, we provide qualitative results on uncertainty estimation, and to complement them, we report here the quantitative results. Results are shown in Table 5, which demonstrate that our FUML is more effective than ETMC and ECML in uncertainty estimation.

*Table 5.* Quantitative results of uncertainty estimation. The normal test sets serve as in-distribution, while the conflicting test sets serve as out-of-distribution. The evaluation metric is FPR95, where lower values indicate better performance. The best results are highlighted in **boldface**.

| METHOD / DATASET | ETMC | ECML | FUML (OURS) |
|---|---|---|---|
| FASHION | 0.930 | 0.967 | **0.510** |
| LANDUSE | 0.964 | 0.945 | **0.886** |

## C.3. Additional Multi-view Fusion Effectiveness Analysis

In this section, we present additional experimental results for analyzing the effectiveness of multi-view fusion. As shown in Figure 7, the prediction error of the multi-view approach is consistently lower than that of any single view in the proposed method. These results confirm that our method effectively reduces prediction error by integrating multiple views to achieve more accurate results, aligning with the observations reported in Section 4.4.

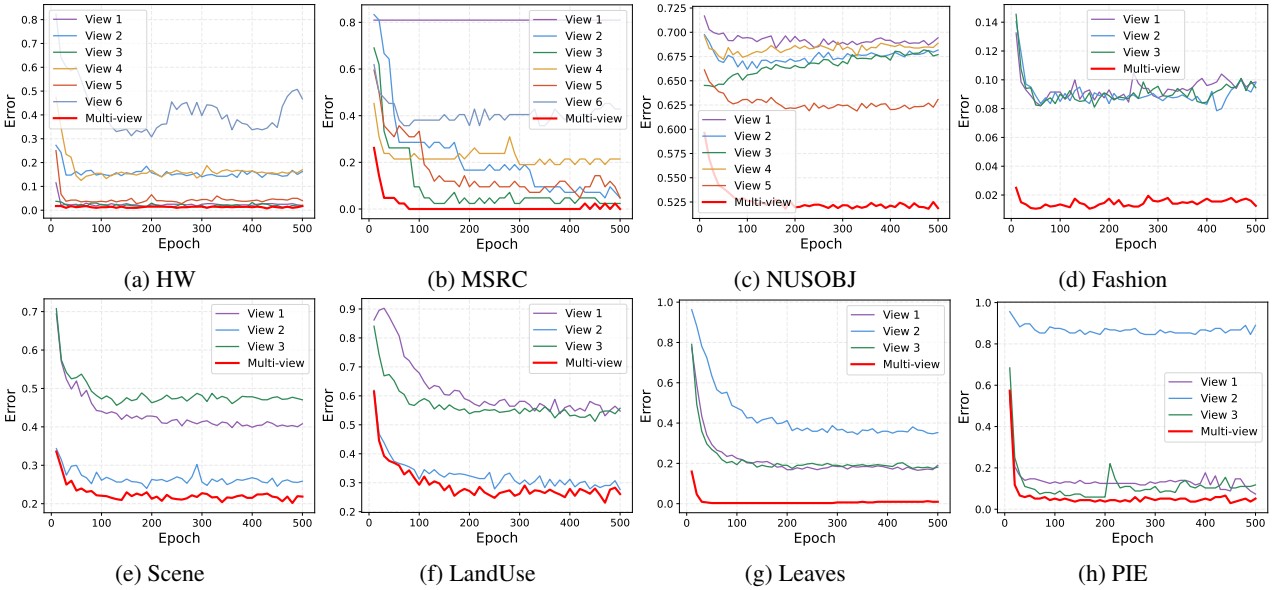

*Figure 7.* Prediction error with different epochs.

## C.4. Additional Ablation Study

In this section, we report the additional ablation experimental results: **(1)** Results in Table 6 show that all components contribute positively to performance. Compared to Concat (i.e., concatenate all view features) and Avg (i.e., $\mathbf{m}_i^a = (\sum_{v=1}^{V} \mathbf{m}_i^v)/V$), DRF is more effective on conflicting test sets than on normal test sets. **(2)** To evaluate the role of necessity, we removed the necessity in FUML and only used conflicts in the fusion process. Results in Table 7 prove that necessity can't be removed. **(3)** To further evaluate the role of uncertainty and conflict in DRF, we conduct more detailed ablation experiments. Results in Table 8 show that removing uncertainty ($u$) or conflict ($c$) in DRF leads to performance degradation, indicating the effectiveness of considering both uncertainty and conflict.

*Table 6.* Classification accuracy (ACC), Precision (Prec.), and F-score of FUML with different combination rules on the normal and conflicting test sets. All metrics are expressed as percentages (%). The best results are highlighted in **boldface**.

| $\mathcal{L}_{ccl}^a$ | $\mathcal{L}_{ccl}^v$ | RULE | FASHION (NORMAL) ACC↑ | PREC.↑ | F-SCORE↑ | LANDUSE (NORMAL) ACC↑ | PREC.↑ | F-SCORE↑ | FASHION (CONFLICTING) ACC↑ | PREC.↑ | F-SCORE↑ | LANDUSE (CONFLICTING) ACC↑ | PREC.↑ | F-SCORE↑ |
|---|---|---|---|---|---|---|---|---|---|---|---|---|---|---|
| √ | × | DRF | 98.66 | 98.66 | 98.48 | 76.29 | 76.89 | 76.09 | 96.33 | 96.31 | 96.30 | 68.00 | 68.72 | 67.53 |
| × | √ | DRF | 98.73 | 98.73 | 98.73 | 75.71 | 76.14 | 75.24 | 96.32 | 96.32 | 96.30 | 67.76 | 68.41 | 67.33 |
| √ | × | CONCAT | 95.71 | 95.71 | 95.70 | 72.26 | 72.44 | 71.55 | 88.45 | 88.28 | 88.75 | 59.69 | 60.54 | 58.54 |
| √ | √ | AVG | 98.50 | 98.51 | 98.51 | 76.19 | 77.14 | 76.07 | 96.15 | 96.13 | 96.13 | 67.71 | 68.22 | 67.42 |
| √ | √ | DRF | **98.96** | **98.97** | **98.96** | **76.71** | **77.57** | **76.48** | **96.68** | **96.68** | **96.68** | **69.14** | **70.21** | **69.19** |

*Table 7.* Classification accuracy (%) on the normal and conflicting test set of the Fashion and LandUse datasets. The means and standard deviations over ten runs are reported. The best results are highlighted in **boldface**.

| DATASET / METHOD | FASHION (NORMAL) | LANDUSE (NORMAL) | FASHION (CONFLICTING) | LANDUSE (CONFLICTING) |
|---|---|---|---|---|
| W/O NECESSITY | 97.70±0.41 | 44.64±3.65 | 94.91±0.47 | 39.00±3.29 |
| OURS | **98.96±0.25** | **76.71±0.46** | **96.68±0.32** | **69.14±2.43** |

*Table 8.* Classification accuracy (%) on the conflicting test set of the Fashion and LandUse datasets. The means and standard deviations over ten runs are reported. The best results are highlighted in **boldface**.

| DATASET / METHOD | FASHION | LANDUSE |
|---|---|---|
| AVG | 96.15±0.22 | 67.71±2.30 |
| $1 - u_i^v$ | 96.27±0.22 | 68.14±2.47 |
| $1 - c_i^v$ | 96.43±0.32 | 68.50±2.02 |
| OURS | **96.68±0.32** | **69.14±2.28** |

## C.5. Identification of Out-of-distribution

This section presents out-of-distribution (OOD) detection results for FUML across all datasets. To validate the effectiveness of our FUML as a trusted method in data noise identification, we add Gaussian noise with fixed standard deviation ($\delta = 0.1, 0.5, 1.0$) to all of the test samples in the test sets, creating OOD samples. In contrast, the remaining data were treated as in-distribution (ID) samples. The results are shown in Figure 8, indicating that ID samples exhibit consistently lower uncertainty relative to OOD samples across all eight datasets. Moreover, OOD samples with greater Gaussian noise deviation generally demonstrate higher uncertainty. Additionally, datasets with higher prediction accuracy (e.g., HW, MSRC, Fashion, Leaves, and PIE) exhibit lower overall uncertainty, whereas datasets with lower prediction accuracy, such as NUSOJB, Scene, and LandUse, tend to display higher uncertainty. These results demonstrate that FUML effectively measures uncertainty, thereby ensuring the reliability of the model's decisions.

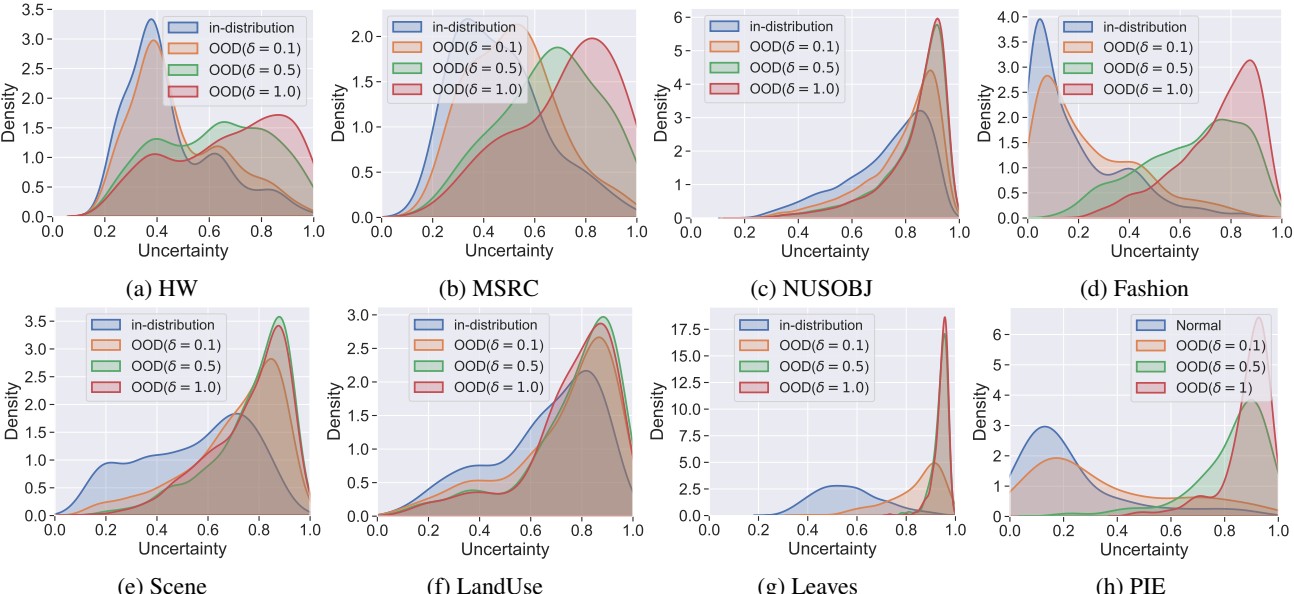

*Figure 8.* Density of uncertainty.

## C.6. Parameter Analysis

To investigate the parameter sensitivity of our method, we plot the accuracy, precision, and F-score of multi-view classification versus different $p$ on the test sets of all the datasets as shown in Figure 9. The results show that in all datasets, the accuracy, precision, and F-score all increase first and then decrease with the increase of $p$. In general, performance is best when $p$ is within 2-5. In this paper, we set $p$ to 3 on all datasets. If we adjust the $p$ value in our model, we believe our FUML will improve over what is reported in the Table 2 and Table 3.

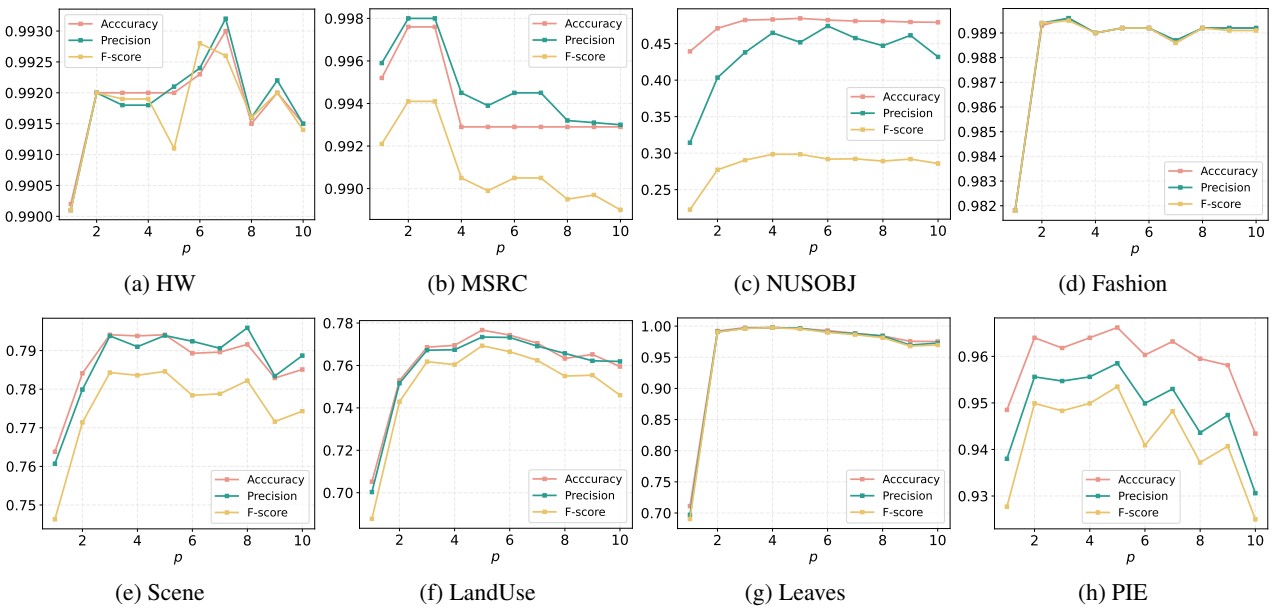

*Figure 9.* The influence of $p$.

## C.7. Challenges of DST in ECML

In evidential deep learning, the outputs of the classification neural network are modeled as evidence. Here, we define the evidence vector for $v$-th view as $\mathbf{e}^v = [e_1^v, ..., e_K^v]$. The parameter $\alpha_k^v = [\alpha_1^v, ..., \alpha_K^v]$ of the Dirichlet distribution is induced from evidence, i.e., $\alpha_k^v = e_k^v + 1$. Then, the belief mass $b_k^v$ and the uncertainty $u^v$ are computed as

$$b_k^v = \frac{e_k^v}{S^v} = \frac{\alpha_k^v - 1}{S^v}, \text{and } u^v = \frac{K}{S^v}, \tag{27}$$

where $S^v = \sum_{i=1}^{K}(e_i^v +) = \sum_{i=1}^{K} \alpha_i^v$ is the Dirichlet strength. Next, we analyze why the decision-level fusion in ECML (Xu et al., 2024a) is order-dependent. In the ECML framework, it is established that fusing two opinions ($\boldsymbol{w} = (\boldsymbol{b}, \boldsymbol{u}, \boldsymbol{a})$) could be mathematically represented as averaging two pieces of evidence, using the following formula:

$$\boldsymbol{w} = \boldsymbol{w}^1 \underline{\lozenge} \, \boldsymbol{w}^2 \underline{\lozenge} \, ... \underline{\lozenge} \, \boldsymbol{w}^V. \tag{28}$$

This formula is implemented in the code snippet:

```
evidence_a = evidences[0]
for v in range(num_views):
    evidence_a = (evidences[i] + evidence_a) / 2
```

ECML essentially applies the D-S combination rule by sequentially merging evidence from different views. However, it has a significant limitation: its decision-level fusion is order-dependent, where the later decision (here referring to evidence) strongly influences the final decision. As a result, ECML is sensitive to the order of conflicting views, making the final decision less reliable when contradictions arise between views. The proposed FUML uses uncertainty and conflict to calculate weights and simultaneously fuses the decision of multiple views at the decision-level fusion, thus avoiding this problem. The results are shown in the Table 12, confirming this.

*Table 9.* We add Gaussian noise on random 50% modalities of 50% random samples in the test set and $\delta$ presents the standard deviation. The means and standard deviations of classification accuracies (%) over ten runs are reported. The best results are highlighted in **boldface**.

| DATASET | METHOD | $\delta = 0$ | $\delta = 1$ | $\delta = 5$ | $\delta = 10$ |
|---------|--------|--------------|--------------|--------------|---------------|
| HW | DCP-CG | 99.00±0.47 | **98.02±0.52** | **96.40±1.28** | 93.40±1.12 |
| | QMF | 98.72±0.48 | 94.85±0.99 | 77.32±1.46 | 71.23±1.56 |
| | PDF | 98.40±0.37 | 95.40±0.98 | 78.92±1.65 | 73.55±1.68 |
| | UIMC | 98.25±0.00 | 97.22±0.13 | 94.70±0.24 | **93.42±0.23** |
| | ECML | 98.72±0.39 | 89.55±1.75 | 73.03±1.51 | 70.18±1.21 |
| | CCML | 97.60±0.63 | 94.05±1.32 | 72.17±1.51 | 66.85±1.80 |
| | OURS | **99.20±0.36** | 96.78±0.82 | 92.38±1.53 | 91.72±1.72 |
| MSRC | DCP-CG | 95.24±3.69 | 79.05±2.78 | 75.24±3.23 | 74.14±3.43 |
| | QMF | 97.86±1.28 | 93.57±2.62 | 73.33±0.60 | 68.57±5.19 |
| | PDF | 97.14±1.78 | 96.43±2.44 | 80.71±3.27 | 73.81±4.64 |
| | UIMC | 98.81±1.19 | 97.21±1.29 | 94.71±1.78 | 93.29±1.90 |
| | ECML | 94.05±1.60 | 83.33±3.53 | 68.33±3.38 | 65.95±3.85 |
| | CCML | 96.90±2.39 | 91.90±1.90 | 73.33±7.59 | 66.43±6.16 |
| | OURS | **99.76±0.75** | **98.10±2.33** | **95.00±2.49** | **94.29±3.05** |
| NUSOBJ | DCP-CG | 43.65±1.10 | 29.11±1.28 | 28.73±1.13 | 28.67±1.12 |
| | QMF | 45.41±0.43 | 32.94±0.31 | 30.62±0.39 | 30.25±0.43 |
| | PDF | 46.78±0.33 | 34.15±0.32 | 31.94±0.33 | 31.69±0.35 |
| | UIMC | 43.42±0.12 | 40.97±0.12 | 36.67±0.13 | 35.85±0.19 |
| | ECML | 42.62±0.42 | 31.59±0.65 | 30.33±0.53 | 29.91±0.48 |
| | CCML | 41.43±0.71 | 29.96±0.87 | 28.01±0.67 | 27.75±0.65 |
| | OURS | **48.23±0.42** | **41.49±0.49** | **41.20±0.54** | **41.16±0.55** |
| FASHION | DCP-CG | 98.11±0.23 | 86.77±2.10 | 82.45±2.67 | 82.24±2.66 |
| | QMF | 98.93±0.32 | 93.05±0.48 | 71.88±0.72 | 69.94±0.69 |
| | PDF | 98.95±0.19 | 93.16±0.59 | 79.83±0.81 | 73.28±0.70 |
| | UIMC | 98.13±0.13 | **95.57±0.12** | **93.04±0.21** | **92.49±0.18** |
| | ECML | 97.93±0.35 | 91.00±0.70 | 76.76±0.74 | 73.16±0.67 |
| | CCML | 95.16±0.41 | 91.39±0.43 | 78.22±0.84 | 72.49±0.87 |
| | OURS | **98.96±0.25** | 95.22±0.32 | 90.66±0.85 | 89.82±0.91 |
| SCENE | DCP-CG | 77.79±1.73 | 62.52±1.22 | 55.50±0.60 | 54.89±0.52 |
| | QMF | 68.58±1.49 | 52.68±1.63 | 48.46±1.57 | 47.90±1.89 |
| | PDF | 70.25±1.21 | 55.25±1.39 | 50.49±1.54 | 49.78±1.61 |
| | UIMC | 77.70±0.00 | **72.32±0.56** | 66.35±0.65 | 65.63±0.72 |
| | ECML | 76.19±0.12 | 58.08±1.34 | 57.66±1.86 | 57.09±1.91 |
| | CCML | 73.87±1.83 | 51.43±2.10 | 47.87±1.60 | 47.50±1.66 |
| | OURS | **79.41±1.34** | 70.14±2.27 | **69.11±2.24** | **68.97±2.25** |
| LANDUSE | DCP-CG | 75.74±0.98 | 55.05±1.66 | 53.76±1.08 | 53.57±0.93 |
| | QMF | 47.86±2.55 | 35.36±2.65 | 33.38±2.56 | 33.43±2.75 |
| | PDF | 45.17±2.66 | 36.21±2.17 | 34.64±1.74 | 34.31±1.71 |
| | UIMC | 57.95±0.61 | 53.26±0.56 | 47.93±0.59 | 47.10±0.55 |
| | ECML | 60.10±2.01 | 39.81±1.27 | 37.61±2.11 | 37.14±2.13 |
| | CCML | 60.86±1.93 | 45.76±1.75 | 42.95±1.63 | 42.71±1.72 |
| | OURS | **76.71±0.46** | **63.86±1.55** | **63.50±1.41** | **63.48±1.47** |
| LEAVES | DCP-CG | 98.19±0.46 | 65.38±1.22 | 65.31±1.20 | 65.21±1.31 |
| | QMF | 95.69±1.25 | 74.16±2.08 | 64.75±2.06 | 64.06±2.20 |
| | PDF | 98.03±0.71 | 77.62±2.42 | 68.34±1.85 | 66.94±2.07 |
| | UIMC | 95.31±0.71 | 91.66±0.97 | 82.94±0.97 | 82.50±0.93 |
| | ECML | 92.53±1.94 | 72.94±1.89 | 65.78±1.99 | 64.84±2.18 |
| | CCML | 97.72±0.92 | 60.03±1.99 | 56.91±2.34 | 56.69±2.50 |
| | OURS | **99.78±0.27** | **93.69±1.17** | **92.94±1.36** | **92.84±1.40** |
| PIE | DCP-CG | 90.59±1.99 | 61.18±3.40 | 61.18±3.40 | 61.18±3.40 |
| | QMF | 92.06±1.64 | 75.51±2.61 | 63.64±3.05 | 62.79±3.11 |
| | PDF | 92.57±1.66 | 77.35±3.29 | 65.81±3.67 | 63.90±2.95 |
| | UIMC | 91.69±2.16 | 89.25±2.22 | 84.81±2.03 | 83.85±2.01 |
| | ECML | 94.71±0.02 | 74.63±3.21 | 64.85±3.48 | 63.31±3.52 |
| | CCML | 93.97±1.67 | 75.44±3.53 | 63.97±2.90 | 62.57±3.06 |
| | OURS | **96.18±1.24** | **89.41±1.68** | **88.38±2.07** | **88.38±2.02** |

*Table 10.* We add Gaussian noise with standard deviation 1 on 50% modalities in different proportions of the test sets. The means and standard deviations of classification accuracies (%) over ten runs are reported. The best results are highlighted in **boldface**.

| DATASET | METHOD | 0% | 10% | 20% | 30% | 40% | 50% |
|---------|--------|-----|------|------|------|------|------|
| HW | DCP-CG | 99.00±0.47 | 98.45±0.37 | **98.45±0.37** | **98.25±0.34** | **98.15±0.45** | **98.02±0.52** |
| | QMF | 98.72±0.48 | 97.58±0.79 | 96.75±0.72 | 96.20±0.84 | 95.67±0.90 | 94.85±0.99 |
| | PDF | 98.40±0.37 | 97.82±0.55 | 97.22±0.64 | 96.60±0.90 | 96.02±0.97 | 95.40±0.98 |
| | UIMC | 98.25±0.00 | 98.15±0.12 | 97.95±0.12 | 97.95±0.12 | 97.92±0.12 | 97.22±0.13 |
| | ECML | 98.72±0.39 | 96.18±0.58 | 93.92±0.81 | 92.30±0.76 | 91.05±1.09 | 89.55±1.75 |
| | CCML | 97.60±0.63 | 96.85±0.71 | 96.05±0.99 | 95.35±1.02 | 94.82±1.28 | 94.05±1.32 |
| | OURS | **99.20±0.36** | **98.62±0.45** | 98.15±0.53 | 97.82±0.63 | 97.38±0.60 | 96.78±0.82 |
| MSRC | DCP-CG | 95.24±3.69 | 91.43±2.43 | 88.10±2.61 | 84.29±2.86 | 82.86±3.16 | 79.05±2.78 |
| | QMF | 97.86±1.28 | 96.90±1.86 | 95.95±2.14 | 95.24±2.13 | 94.76±2.08 | 93.57±2.62 |
| | PDF | 97.14±1.78 | 96.90±1.86 | 96.90±2.39 | 96.90±2.14 | 96.9±2.14 | 96.43±2.44 |
| | UIMC | 98.81±1.19 | 98.05±1.17 | 98.05±1.17 | 97.81±1.19 | 97.81±1.19 | 97.21±1.29 |
| | ECML | 94.05±1.60 | 91.90±2.43 | 89.52±2.86 | 88.10±3.53 | 86.43±3.38 | 83.33±3.53 |
| | CCML | 96.90±2.39 | 94.05±3.06 | 93.57±3.2 | 93.57±3.2 | 92.62±2.49 | 91.90±1.90 |
| | OURS | **99.76±0.75** | **99.76±0.71** | **99.29±1.09** | **98.81±1.19** | **98.81±1.19** | **98.10±2.33** |
| NUSOBJ | DCP-CG | 43.65±1.10 | 41.09±0.75 | 38.22±0.56 | 35.43±0.52 | 32.62±0.56 | 29.11±1.28 |
| | QMF | 45.41±0.43 | 42.71±0.42 | 40.12±0.43 | 37.27±0.31 | 34.32±0.38 | 32.94±0.31 |
| | PDF | 46.78±0.33 | 45.94±0.49 | 43.08±0.48 | 40.09±0.23 | 37.10±0.26 | 34.15±0.32 |
| | UIMC | 43.42±0.12 | 42.95±0.13 | 42.33±0.15 | 41.81±0.14 | 41.27±0.17 | 40.97±0.12 |
| | ECML | 42.62±0.42 | 40.62±0.89 | 38.50±0.84 | 37.09±0.66 | 35.06±0.82 | 31.59±0.65 |
| | CCML | 41.43±0.71 | 38.76±0.64 | 36.57±0.79 | 34.33±0.67 | 32.07±0.77 | 29.96±0.87 |
| | OURS | **48.23±0.42** | **46.66±0.75** | **45.39±0.68** | **44.10±0.58** | **42.78±0.56** | **41.49±0.49** |
| FASHION | DCP-CG | 98.11±0.23 | 95.42±0.55 | 93.33±0.89 | 91.09±1.29 | 88.87±1.75 | 86.77±2.10 |
| | QMF | 98.93±0.32 | 96.84±0.51 | 95.67±0.56 | 94.71±0.45 | 93.78±0.51 | 93.05±0.48 |
| | PDF | 98.95±0.19 | 97.57±0.36 | 96.64±0.43 | 95.82±0.46 | 95.14±0.49 | 93.16±0.59 |
| | UIMC | 98.13±0.13 | 97.57±0.12 | 97.42±0.12 | **97.20±0.10** | **96.97±0.12** | **95.57±0.12** |
| | ECML | 97.93±0.35 | 96.44±0.42 | 94.91±0.48 | 93.48±0.51 | 92.14±0.59 | 91.00±0.70 |
| | CCML | 95.16±0.41 | 94.47±0.55 | 93.58±0.48 | 92.78±0.52 | 92.07±0.48 | 91.39±0.43 |
| | OURS | **98.96±0.25** | **98.22±0.27** | **97.43±0.28** | 96.75±0.33 | 95.94±0.28 | 95.22±0.32 |
| SCENE | DCP-CG | 77.79±1.73 | 74.85±0.83 | 71.37±0.28 | 68.38±0.52 | 65.66±0.96 | 62.52±1.22 |
| | QMF | 68.58±1.49 | 65.32±1.60 | 62.15±1.44 | 59.31±1.25 | 55.89±1.45 | 52.68±1.63 |
| | PDF | 70.25±1.21 | 67.28±1.26 | 64.37±1.20 | 61.46±0.94 | 58.25±0.95 | 55.25±1.39 |
| | UIMC | 77.70±0.00 | 76.10±0.54 | 74.95±0.56 | 73.43±0.54 | **72.93±0.54** | **72.32±0.56** |
| | ECML | 76.19±0.12 | 69.94±1.51 | 66.73±1.46 | 63.92±1.44 | 60.81±1.50 | 58.08±1.34 |
| | CCML | 73.87±1.83 | 64.56±1.64 | 61.19±1.49 | 58.06±1.67 | 54.65±1.78 | 51.43±2.10 |
| | OURS | **79.41±1.34** | **77.53±1.83** | **75.47±1.95** | **73.77±2.13** | 72.04±2.09 | 70.14±2.27 |
| LANDUSE | DCP-CG | 75.74±0.98 | 72.29±1.95 | 68.10±1.24 | 64.29±1.33 | 60.00±0.96 | 55.05±1.66 |
| | QMF | 47.86±2.55 | 45.31±2.75 | 42.83±2.60 | 40.69±2.63 | 38.12±2.67 | 35.36±2.65 |
| | PDF | 45.17±2.66 | 44.76±2.64 | 42.79±2.54 | 40.69±2.14 | 38.62±2.13 | 36.21±2.17 |
| | UIMC | 57.95±0.61 | 57.05±0.47 | 55.81±0.47 | 55.52±0.44 | 54.71±0.44 | 53.26±0.56 |
| | ECML | 60.10±2.01 | 49.62±2.36 | 46.90±2.23 | 45.10±1.96 | 42.62±1.73 | 39.81±1.27 |
| | CCML | 60.86±1.93 | 57.67±1.89 | 54.86±2.45 | 52.05±2.49 | 48.67±2.36 | 45.76±1.75 |
| | OURS | **76.71±0.46** | **73.21±1.83** | **71.07±1.83** | **68.88±1.71** | **66.38±1.83** | **63.86±1.55** |
| LEAVES | DCP-CG | 98.19±0.46 | 91.06±0.70 | 84.25±1.06 | 77.81±1.06 | 71.56±1.27 | 65.38±1.22 |
| | QMF | 95.69±1.25 | 91.31±1.20 | 87.03±1.72 | 82.37±1.97 | 78.34±2.43 | 74.16±2.08 |
| | PDF | 98.03±0.71 | 94.09±1.13 | 89.91±1.58 | 85.62±2.22 | 81.41±2.75 | 77.62±2.42 |
| | UIMC | 95.31±0.71 | 94.41±0.83 | 93.81±0.80 | 93.03±0.77 | 92.03±0.82 | 91.66±0.97 |
| | ECML | 92.53±1.94 | 88.69±1.18 | 84.03±1.65 | 79.62±1.30 | 75.34±1.89 | 72.94±1.89 |
| | CCML | 97.72±0.92 | 79.66±1.84 | 74.66±1.57 | 69.72±1.37 | 65.13±1.82 | 60.03±1.99 |
| | OURS | **99.78±0.27** | **98.59±0.64** | **97.56±0.80** | **96.12±1.02** | **94.88±1.24** | **93.69±1.17** |
| PIE | DCP-CG | 90.59±1.99 | 83.24±3.73 | 77.50±3.34 | 71.32±2.98 | 66.18±2.94 | 61.18±3.40 |
| | QMF | 92.06±1.64 | 88.75±1.95 | 85.22±2.04 | 81.62±2.42 | 78.46±3.19 | 75.51±2.61 |
| | PDF | 92.57±1.66 | 89.71±1.83 | 86.32±2.19 | 83.38±2.55 | 80.07±3.13 | 77.35±3.29 |
| | UIMC | 91.69±2.16 | 90.99±2.16 | 90.49±1.93 | 90.40±2.08 | **90.18±2.25** | 89.25±2.22 |
| | ECML | 94.71±0.02 | 87.06±2.77 | 84.49±2.64 | 82.12±2.33 | 79.49±2.43 | 76.63±3.21 |
| | CCML | 93.97±1.67 | 88.53±2.08 | 84.56±1.89 | 81.62±2.85 | 78.31±3.48 | 75.44±3.53 |
| | OURS | **96.18±1.24** | **93.97±1.27** | **92.72±1.59** | **91.32±1.57** | 90.07±1.84 | **89.41±1.68** |

*Table 11.* We add unaligned views on 50% modalities of the test sets in different proportions. The means and standard deviations of classification accuracies (%) over ten runs are reported. The best results are highlighted in **boldface**.

| DATASET | METHOD | 0% | 10% | 20% | 30% | 40% | 50% |
|---|---|---|---|---|---|---|---|
| HW | DCP-CG | 99.00±0.47 | 98.45±0.37 | 98.35±0.34 | 98.30±0.29 | 98.05±0.33 | 97.95±0.43 |
| | QMF | 98.72±0.48 | 98.42±0.46 | 98.38±0.44 | 98.30±0.42 | 98.12±0.46 | 97.78±0.59 |
| | PDF | 98.40±0.37 | 97.50±0.43 | 96.72±0.55 | 95.55±0.58 | 94.50±0.80 | 93.60±0.71 |
| | UIMC | 98.25±0.00 | 98.20±0.19 | 97.88±0.13 | 97.72±0.11 | 97.72±0.11 | 97.72±0.11 |
| | ECML | 98.72±0.39 | 97.48±0.71 | 96.92±0.81 | 96.02±0.72 | 95.45±0.95 | 94.60±0.77 |
| | CCML | 97.60±0.63 | 96.23±0.75 | 95.40±1.02 | 94.40±1.17 | 93.40±1.41 | 92.25±1.53 |
| | OURS | **99.20±0.36** | **99.13±0.28** | **99.10±0.28** | **99.05±0.33** | **99.05±0.31** | **99.10±0.32** |
| MSRC | DCP-CG | 95.24±3.69 | 94.71±3.50 | 94.71±2.78 | 93.29±3.56 | 93.29±3.56 | 92.33±4.10 |
| | QMF | 97.86±1.28 | 97.14±1.78 | 96.67±2.18 | 95.95±2.14 | 95.48±1.98 | 95.00±2.70 |
| | PDF | 97.14±1.78 | 95.71±1.78 | 94.76±2.08 | 92.86±2.61 | 92.38±3.16 | 91.43±3.23 |
| | UIMC | 98.81±1.19 | 98.05±1.17 | 98.05±1.17 | 96.57±1.17 | 96.57±1.17 | 91.90±1.17 |
| | ECML | 94.05±1.60 | 93.57±3.02 | 93.33±2.78 | 92.62±2.70 | 90.95±2.97 | 89.29±2.44 |
| | CCML | 96.90±2.39 | 94.29±2.86 | 94.05±3.24 | 92.38±3.66 | 91.90±3.40 | 91.67±3.41 |
| | OURS | **99.76±0.75** | **99.76±0.71** | **99.76±0.71** | **99.76±0.71** | **99.76±0.71** | **99.76±0.71** |
| NUSOBJ | DCP-CG | 43.65±1.10 | 43.22±0.95 | 42.56±1.05 | 41.96±1.01 | 41.42±1.04 | 40.86±1.08 |
| | QMF | 45.41±0.43 | 45.22±0.40 | 45.15±0.43 | 44.94±0.55 | 44.76±0.58 | 44.72±0.59 |
| | PDF | 46.78±0.33 | 46.50±0.29 | 46.24±0.44 | 45.92±0.46 | 45.58±0.49 | 45.35±0.46 |
| | UIMC | 43.42±0.12 | 43.14±0.15 | 42.76±0.19 | 42.62±0.12 | 42.21±0.17 | 41.78±0.14 |
| | ECML | 42.62±0.42 | 41.92±0.59 | 41.29±0.57 | 41.03±0.58 | 40.71±0.58 | 40.53±0.52 |
| | CCML | 41.43±0.71 | 40.40±0.42 | 39.98±0.55 | 39.54±0.57 | 38.92±0.64 | 38.43±0.70 |
| | OURS | **48.23±0.42** | **47.78±0.58** | **47.62±0.50** | **47.40±0.54** | **47.07±0.54** | **46.86±0.49** |
| FASHION | DCP-CG | 98.11±0.23 | 96.00±1.15 | 94.71±1.56 | 93.01±2.06 | 91.33±2.58 | 89.93±3.00 |
| | QMF | 98.93±0.32 | 97.77±0.32 | 96.63±0.24 | 95.48±0.33 | 94.34±0.42 | 93.16±0.39 |
| | PDF | 98.95±0.19 | 97.17±0.23 | 95.50±0.36 | 93.77±0.54 | 91.84±0.92 | 89.68±1.03 |
| | UIMC | 98.13±0.13 | 96.12±0.19 | 93.75±0.23 | 91.85±0.21 | 89.90±0.27 | 88.12±0.33 |
| | ECML | 97.93±0.35 | 94.43±0.32 | 91.20±0.39 | 87.91±0.51 | 84.58±0.60 | 81.49±0.68 |
| | CCML | 95.16±0.41 | 92.55±0.32 | 89.76±0.24 | 86.81±0.62 | 84.08±0.85 | 81.45±1.07 |
| | OURS | **98.96±0.25** | **98.57±0.21** | **98.18±0.24** | **97.78±0.19** | **97.32±0.21** | **96.88±0.23** |
| SCENE | DCP-CG | 77.79±1.73 | 74.27±0.64 | 72.33±0.69 | 69.90±1.29 | 67.56±1.56 | 65.04±1.51 |
| | QMF | 68.58±1.49 | 66.83±1.30 | 65.22±1.11 | 63.57±1.01 | 61.67±0.97 | 60.07±0.99 |
| | PDF | 70.25±1.21 | 67.87±1.17 | 65.54±1.12 | 63.14±0.98 | 60.66±1.00 | 58.13±1.36 |
| | UIMC | 77.70±0.00 | 74.86±0.46 | 72.85±0.45 | 70.79±0.45 | 68.74±0.43 | 66.12±0.45 |
| | ECML | 76.19±0.12 | 71.43±1.31 | 69.86±1.16 | 68.26±1.17 | 66.73±1.16 | 65.37±1.22 |
| | CCML | 73.87±1.83 | 65.60±1.60 | 63.31±1.55 | 61.43±1.51 | 59.16±1.57 | 57.05±1.53 |
| | OURS | **79.41±1.34** | **78.19±1.43** | **76.98±1.52** | **75.80±1.39** | **74.68±1.30** | **73.99±1.15** |
| LANDUSE | DCP-CG | 75.74±0.98 | 73.33±2.03 | 69.57±1.12 | 66.71±1.56 | 63.00±1.76 | 59.95±1.69 |
| | QMF | 47.86±2.55 | 46.60±2.61 | 45.10±2.62 | 44.02±2.56 | 42.67±2.18 | 41.38±2.12 |
| | PDF | 45.17±2.66 | 44.50±2.49 | 42.62±2.33 | 40.64±2.31 | 38.88±1.95 | 37.07±2.12 |
| | UIMC | 57.95±0.61 | 56.40±0.58 | 53.45±0.56 | 51.83±0.41 | 49.36±0.41 | 47.30±0.63 |
| | ECML | 60.10±2.01 | 52.17±1.18 | 50.79±1.58 | 49.62±1.69 | 48.00±1.38 | 46.76±1.89 |
| | CCML | 60.86±1.93 | 59.29±1.87 | 57.80±2.29 | 55.48±2.96 | 54.17±2.70 | 53.21±3.07 |
| | OURS | **76.71±0.46** | **74.52±1.91** | **73.38±1.81** | **72.10±1.74** | **70.98±1.61** | **69.93±1.47** |
| LEAVES | DCP-CG | 98.19±0.46 | 95.06±1.69 | 91.19±1.30 | 88.00±1.84 | 84.69±2.19 | 81.56±2.01 |
| | QMF | 95.69±1.25 | 91.50±1.32 | 87.66±1.71 | 83.25±1.81 | 79.06±2.11 | 75.38±2.24 |
| | PDF | 98.03±0.71 | 92.78±1.21 | 87.75±1.36 | 82.34±1.02 | 77.56±0.94 | 72.56±0.84 |
| | UIMC | 95.31±0.71 | 89.72±0.65 | 86.47±0.74 | 83.06±0.87 | 80.41±0.94 | 77.09±0.97 |
| | ECML | 92.53±1.94 | 89.84±1.95 | 86.41±1.84 | 83.09±1.82 | 79.62±2.09 | 76.19±2.25 |
| | CCML | 97.72±0.92 | 81.09±2.22 | 78.06±2.01 | 74.59±2.01 | 71.34±1.69 | 68.12±1.71 |
| | OURS | **99.78±0.27** | **98.94±0.72** | **97.84±1.14** | **96.72±1.55** | **95.81±1.51** | **94.69±1.26** |
| PIE | DCP-CG | 90.59±1.99 | 85.15±3.06 | 81.03±2.05 | 76.76±1.58 | 72.65±2.43 | 68.68±1.36 |
| | QMF | 92.06±1.64 | 90.51±1.81 | 88.82±2.37 | 86.40±2.68 | 84.34±2.61 | 81.99±2.98 |
| | PDF | 92.57±1.66 | 89.19±1.44 | 85.96±2.07 | 82.06±1.89 | 79.12±2.44 | 75.00±2.81 |
| | UIMC | 91.69±2.16 | 85.96±2.40 | 82.72±2.33 | 77.06±2.79 | 73.09±2.72 | 68.24±2.82 |
| | ECML | 94.71±0.02 | 88.75±2.51 | 86.69±2.47 | 84.56±2.28 | 82.50±1.97 | 80.15±2.55 |
| | CCML | 93.97±1.67 | 90.22±1.86 | 88.24±1.64 | 86.03±1.77 | 84.26±1.62 | 82.65±1.95 |
| | OURS | **96.18±1.24** | **94.93±1.37** | **93.38±1.95** | **91.47±1.92** | **89.85±2.13** | **87.94±2.38** |

*Table 12.* Classification accuracy (%) of ECML and our FUML with varying noise views on the PIE dataset. ECML fuses views from the first view, making it sensitive to the order of the noisy views.

| METHODS | NOISE VIEW | 1+2+3 | 1+3+2 | 2+1+3 | 2+3+1 | 3+1+2 | 3+2+1 | Δ% |
|---|---|---|---|---|---|---|---|---|
| ECML | 1 | 87.13 | 81.69 | 87.13 | 76.91 | 81.76 | 77.21 | 10.22 |
| | 2 | 91.18 | 82.13 | 90.96 | 91.25 | 82.06 | 91.25 | 9.19 |
| | 3 | 86.76 | 88.97 | 86.54 | 90.81 | 88.90 | 91.10 | 4.56 |
| OURS | 1 | | | **93.24** | | | | 0 |
| | 2 | | | **96.18** | | | | 0 |
| | 3 | | | **92.65** | | | | 0 |

## C.8. Conflict Visualization

Figure 10 presents the conflict on the HandWritten dataset with six views. To introduce conflicts, we modify the content in the third view to other categories, resulting in misalignment with the other views. The Figure 10 (a) and (b) depict the conflict of normal and conflicting instances, respectively. The results show that FUML effectively captures and quantifies conflict between views, further validating its reliability.

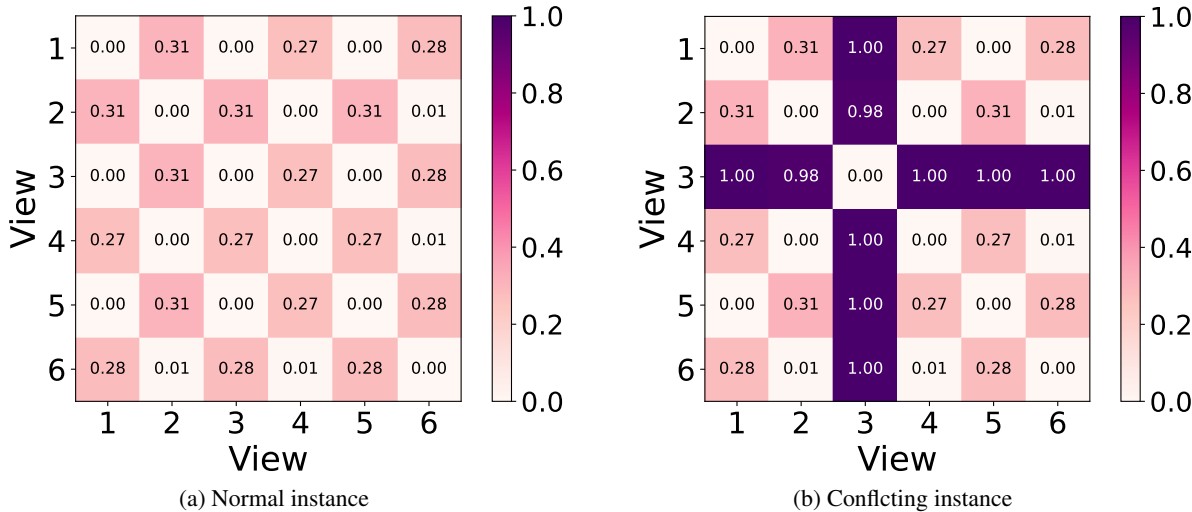

(a) Normal instance        (b) Conflcting instance

*Figure 10.* Conflict visualization.

## C.9. Adversarial Noise Effect Analysis

In this section, we present additional experimental results under adversarial noise attacks. First, to evaluate the performance of our FUML under adversarial noise, we add projected gradient descent (PGD) adversarial noise attacks (Madry et al., 2018) with different maximum perturbation magnitudes (eps) to the test set of the Fashion dataset. The results in Table 13 show FUML's superior resistance to adversarial noise attacks. Second, to further evaluate the effectiveness of the uncertainty estimation mechanism of FUML under adversarial noise, we perform an OOD task on the Fashion dataset, using normal test sets as ID and PGD-attacked sets (eps=0.10) as OOD. Evaluated by FPR95 (lower is better), FUML achieved 0.68, outperforming ETMC and ECML (both 1.00). To sum up, under adversarial noise, the proposed FUML is superior in both classification accuracy and uncertainty estimation.

## D. Limitations

Even though the proposed FUML outperforms existing multi-view classification methods in terms of both performance and reliability, there are still some potential limitations. For instance, FUML's multi-view fusion weights incorporate both uncertainty and conflict through multiplication. Although extensive experiments have demonstrated the effectiveness of this

*Table 13.* Classification accuracy (%) of PDF, ECML, and our FUML under PGD adversarial noise attacks with different eps on the Fashion dataset. The best results are highlighted in **boldface**.

| METHOD \ EPS | 0 | 0.05 | 0.10 |
|:---:|:---:|:---:|:---:|
| PDF | $98.95 \pm 0.19$ | $22.07 \pm 0.90$ | $13.54 \pm 0.89$ |
| ECML | $97.93 \pm 0.35$ | $52.58 \pm 0.51$ | $42.74 \pm 1.35$ |
| OURS | $\mathbf{98.96 \pm 0.25}$ | $\mathbf{94.45 \pm 0.18}$ | $\mathbf{93.40 \pm 0.19}$ |

technique, it lacks corresponding theoretical guarantees. Therefore, it is crucial to explore new multi-view fusion techniques based on uncertainty and conflict from a theoretical standpoint. We hope that our study can serve as a valuable baseline for future research in multi-view learning and uncertainty estimation.

