# OpenReview forum: "Deep Fuzzy Multi-view Learning for Reliable Classification"
_ICML.cc/2025/Conference — ICML 2025 poster_

### Official Review · Reviewer_PsMz · 2025-03-08

**Overall Recommendation:** 4

**Summary:**

This paper introduces FUML, a novel multi-view classification framework using Fuzzy Set Theory to handle conflicting views and improve uncertainty estimation. It employs a Dual-reliable Multi-view Fusion (DRF) strategy and entropy-based uncertainty quantification, achieving robust classification and reliability. Experiments show superior performance over existing methods.

**Claims And Evidence:**

Yes, this paper is supported by clear and convincing evidence.

**Essential References Not Discussed:**

No

**Experimental Designs Or Analyses:**

Yes, I checked the soundness and validity of all experimental designs and analyses in this paper.

**Methods And Evaluation Criteria:**

Yes, the proposed method and evaluation criteria make sense.

**Other Comments Or Suggestions:**

N/A

**Other Strengths And Weaknesses:**

Strengths:

1. This paper effectively clarifies its research motivation with the aid of Figure 1 and introduces a novel approach to integrating fuzzy set theory into multi-view learning.

2. This paper innovatively proposes a TMVC method based on the fuzzy set theory, avoiding the problem of EDL-based TMVC methods being sensitive to conflicts and inaccurate uncertainty estimation. The proposed FUML is simple but effective.

3. The paper has sufficient experiments, for example, comparative experiments with 13 latest baselines (including five untrusted MVC baselines and eight trusted MVC baselines) on eight public datasets. In addition, including the appendix, there are 11 types of experiments, which fully demonstrate the effectiveness of the proposed method.

4. The proposed method achieves significant improvements over the best baseline, especially on the conflict test set. For example, the accuracy is improved by 4.83%, 7.31% and 14.6% on Scene, LandUse and Leaves datasets respectively.

Weaknesses:

1. In the Experimental Setup section, the author mentioned the method of adding conflict samples, but did not provide details on the noise intensity and noise ratio of the added Gaussian noise. Please add relevant description.

2. In the experiments, the authors include some multimodal classification methods such as PDF and QMF in the comparison, while this paper mainly focuses on multi-view classification. Is this fair?

3. In Figure 3, the uncertainty effectiveness Analysis is only performed on Fashion and LandUse dataset which only has three views. I am concerned about the results of other datasets, such as Handwritten or MSRC, which have six views. Because as shown in Table 2, for such datasets with many views, conflicting views often don’t cause a significant drop in accuracy, but should cause an increase in uncertainty.

4. The motivation for introducing necessity is unclear. Nor does authors explain the impact of necessity on performance.

**Questions For Authors:**

1. The FUML method in this paper is mainly used for multi-view classification. Can it be applied to other multimodal classification tasks?If so, what adjustments are needed?

2. Why should necessity be introduced? Corresponding experiments should be supplemented.


Besides, please answer the questions in the weaknesses.

**Relation To Broader Scientific Literature:**

The key contributions of this paper are closely related to the broader scientific literature in multi-view learning, uncertainty estimation and Fuzzy Set Theroy. Specifically, this paper compares prior work in trusted multi-view classification based on Evidential Theory (Including TMC, ETMC, UMIC, and ECML) . Besides, this paper is related to and based on uncertainty estimation andfuzzy theory, and adopts fuzzy modeling, providing new ideas on how to apply fuzzy set theory to multi-view learning.

**Theoretical Claims:**

Yes, I have checked the theoretical part in this paper.

---

> ### Author Rebuttal · Authors · 2025-04-01
>
> We appreciate the identification of our novelty and the positive comments. Below are our point-by-point responses to your concerns:
>
> **Q1: In the Experimental Setup section, the author doesn’t provide details on the noise intensity and noise ratio of the added Gaussian noise.**
>
> **R1**: To create a test set with conflicting instances, following the methodology outlined in (Xu, Cai, et al. "Reliable conflictive multi-view learning."), we apply two transformations: 1)We add Gaussian noise with mean 0 and variance 0.5 to $10\%$ of the instances in the test set. 2)We randomly changed the view information of $50\%$ of the instances in the test set, making the labels of some views inconsistent with the other views. We will clarify this in the next version.
>
> **Q2: In the experiments, the authors include some multimodal classification methods, such as PDF and QMF, in the comparison. Is this fair?**
>
> **R2**: Yes, it is fair. To ensure a fair comparison, we replace the backbone networks of QMF and PDF with the same fully connected layer as FUML while preserving their core models and loss functions. These experimental details can be found in Appendix B.3.
>
> **Q3: Concerns about the uncertainty estimation results of Handwritten or MSRC.**
>
> **R3**: We compare our FUML with ETMC and ECML on the Handwritten and MSRC datasets for the OOD task. The normal test sets serve as in-distribution, while the conflicting test sets serve as OOD. The evaluation metric is FPR95, with lower values indicating better performance. The results below demonstrate the superiority of FUML.
>
> |      | ETMC  | ECML  | FUML(Ours) |
> | ---- | ----- | ----- | ---------- |
> | HW   | 0.920 | 0.950 | 0.845      |
> | MSRC | 0.952 | 0.974 | 0.926      |
>
> **Q4: The motivation for necessity is unclear, and its impact on performance is unexplained. Why introduce it? Relevant experiments are needed.**
>
> **R4**: In Fuzzy Set Theory, necessity is introduced to quantify the certainty that a sample does not belong to other categories, as membership alone cannot capture between-class relationships. To evaluate the role of necessity, we removed the necessity in FUML and only used conflicts in the fusion process. The experimental results in the table below prove that necessity can’t be removed.
>
> |               | Fashion(Normal) | LandUse(Normal) | Fashion (Conflict) | LandUse(Conflict) |
> | ------------- | --------------- | --------------- | ------------------ | ----------------- |
> | w/o Necessity | $97.70\pm0.41$  | $44.64\pm3.65$  | $94.91\pm0.47$     | $39.00\pm3.29$    |
> | Ours          | $98.96\pm0.25$  | $76.71\pm0.46$  | $96.68\pm0.32$     | $69.14\pm2.43$    |
>
> **Q5: Can FUML be applied to multimodal classification tasks? If so, what adjustments are needed?**
>
> **R5**: Yes, our FUML framework exhibits strong generalizability and can be readily adapted to various multimodal classification tasks. The primary adjustments required involve replacing the feature extraction backbones to accommodate different data modalities. Specifically, Vision Transformer (ViT) can be employed for image feature extraction, wav2vec for audio processing, TimeSformer for video analysis, and word2vec for textual feature representation.

---

> > ### Comment · Reviewer_PsMz · 2025-04-03
> >
> > Thanks for your response. All my concerns have been addressed, I will raise my score to 4.

---

> > > ### Author Response · Authors · 2025-04-09
> > >
> > > Thank you for your thoughtful review of our manuscript. We sincerely appreciate the time and effort you dedicated to evaluating our work and providing insightful comments.

---

### Official Review · Reviewer_hton · 2025-03-12

**Overall Recommendation:** 4

**Summary:**

This paper proposes a novel multi-view classification method based on Fuzzy Set Theory, which models classification outputs as fuzzy memberships. After that, authors introduces a category credibility learning loss and a Dual-reliable Fusion (DRF) strategy to manage conflicting views and improve uncertainty estimation over Evidential Deep Learning. Adequate experiments and analysis demonstrate FUML's superior performance in accuracy and reliability over existing methods.

**Claims And Evidence:**

Yes, this paper is supported by clear and convincing evidence.

**Essential References Not Discussed:**

No.

**Experimental Designs Or Analyses:**

Yes, I checked the soundness and validity of all experimental designs and analyses in this paper.

**Methods And Evaluation Criteria:**

Yes, the proposed method and evaluation criteria make sense.

**Other Comments Or Suggestions:**

See the weaknesses.

**Other Strengths And Weaknesses:**

Strengths:

S1. This paper has a clear motivation and a novel perspective and effectively solves the problems of existing EDL-based TMVC methods being sensitive to conflicts and inaccurate uncertainty estimation of conflicting multi-view samples.

S2. This paper is well-written and clearly presented. As far as I know, the use of fuzzy set theory to introduce output uncertainty in multi-view classification is novel.

S3. The proposed Dual-reliable Multi-view Fusion is quite neat and effective.

S4. The author conducted a large number of experiments to verify the effectiveness of their FUML, including both qualitative results (such as Figure 3-10) and quantitative results (such as Table 2-10). In addition, as shown in Table 2, in the case of conflicting multi-view classification, its performance far exceeds the best baselines, especially on the Leaves dataset.


Weaknesses:

W1. There are some grammatical errors in the writing: line 057:  “... have be proposed to ...” should be “... have been proposed to ...”. Line 072: “global conflicts among views overly emphasizes dominant evidence” should be “global conflicts among views overly and emphasizes dominant evidence”. Line 179: " Thirdly the weights ...” should be “ Thirdly, the weights ...”.

W2. The choice of activation function in Eq. 3 warrants further discussion. Specifically, it is unclear whether other activation functions, such as those used in Evidential Deep Learning, could be applied and whether they would offer comparable or improved effectiveness.

W3. In Eq. 8, the exp function is selected as g(). Why not use other monotonically increasing functions, such as ReLU and sigmoid? This deserves further discussion.

W4. In Definition 3.3, the authors used cosine distance to measure the distance between memberships, thereby measuring the conflict between views. Why not use Euclidean distance or Dot Product Similarity?

**Questions For Authors:**

Q1. In this paper, the authors pointed out that the norm is first used, and then the ReLU function is used to model the membership in the fuzzy set theory. Can the ReLU function be replaced by other functions?

Q2. In Eq. 8, the exp function is selected as g(). Why not use other monotonically increasing functions?

Q3. In Definition 3.3, the authors used cosine distance to measure the distance between memberships from different views. Why not use Euclidean distance or Dot Product Similarity? What is the advantage of cosine similarity?

**Relation To Broader Scientific Literature:**

This paper introduces a novel trusted multi-view classification method, which is based on fuzzy set theory and can accurately classify conflicting multi-view instances and precisely estimate classification uncertainty. This work builds on the work of “Trusted Multi-View Classification with  Dynamic Evidential Fusion” which first proposed trusted multi-view classification based on evidential deep learning (EDL), and “Reliable Conflictive Multi-view Learning”, which first proposed conflictive multi-view learning. Its contributions extend beyond multi-view classification, offering new angles for single-view, multimodal classification and out-of-distribution detection.

**Theoretical Claims:**

Yes, I checked the theoretical proof section of this paper in the appendix.

---

> ### Author Rebuttal · Authors · 2025-04-01
>
> We appreciate your detailed comments. We believe the following point-to-point response can address all the concerns:
>
> **Q1: There are some grammatical errors in the writing.**
>
> **R1**: Thanks. We will correct these grammatical errors you raise and carefully review the manuscript to ensure no other grammatical errors remain.
>
> **Q2: Can the ReLU function be replaced by other functions?**
>
> **R2**: No. To map the logits of a neural network as memberships, the process involves two key steps: First, $L_p$-normalization is applied to constrain the logits within the range $[-1, 1]$. Then, a ReLU activation function is employed to further restrict the outputs to the interval $[0, 1]$. The resulting values can be interpreted as memberships for their respective categories. Therefore, both ReLU and $L_p$ normalization are essential parts to qualify membership. If ReLU is replaced by exp or softplus used in the Evidential Deep Learning, the output value will not be in the range of $[0,1]$, which undoubtedly violates the definition of membership.
>
> **Q3**: In Eq. 8, the exp function is selected as g(). Why not use other monotonically increasing functions, such as ReLU and sigmoid?
>
> **R3**: The choice of exp is an empirical result: The performance comparison of ReLU, Sigmoid, and the exp function is shown in the following table. From these results, we can find that exp has the best or second-best performance. Therefore, we leverage exp function as g().
>
> |           | Fashion(Normal) | LandUse(Normal) | Fashion(Conflict) | LandUse(Conflict) |
> | --------- | --------------- | --------------- | ----------------- | ----------------- |
> | ReLU      | $98.81\pm0.26$  | $76.75\pm0.56$  | $96.51\pm0.29$    | $68.80\pm2.49$    |
> | Sigmoid   | $98.75\pm0.27$  | $76.34\pm0.78$  | $96.57\pm0.29$    | $68.94\pm1.89$    |
> | exp(Ours) | $98.96\pm0.25$  | $76.71\pm0.46$  | $96.68\pm0.32$    | $69.14\pm2.43$    |
>
> **Q4**: Why not use Euclidean distance or Dot Product Similarity? What is the advantage of cosine similarity?
>
> **R4**: Compared with Euclidean distance and dot product similarity, cosine similarity only evaluates the consistency of decisions through the angle between memberships, avoiding the sensitivity of Euclidean distance and dot product similarity to vector length, thereby more accurately measuring decision conflicts.

---

### Official Review · Reviewer_8eiq · 2025-03-13

**Overall Recommendation:** 5

**Summary:**

The paper introduces FUML, a novel multi-view classification framework that explicitly addresses the uncertainty caused by conflicting information across views. By leveraging Fuzzy Set Theory, the authors model the outputs of deep classifiers as fuzzy memberships, capturing both possibility and necessity. A tailored loss function (i.e., the category credibility learning loss) is proposed to guide the optimization of these fuzzy outputs. In addition, a Dual-reliable Fusion (DRF) strategy is designed to weight each view based on its estimated uncertainty and inter-view conflict, ensuring that noisy or misaligned views contribute less to the final decision. Extensive experiments on eight public datasets demonstrate that FUML not only improves classification accuracy over 13 state-of-the-art baselines but also provides more robust uncertainty estimates, particularly in scenarios with conflicting view information.

**Claims And Evidence:**

The primary claims of the paper are that:
1.FUML achieves state-of-the-art classification accuracy while providing reliable uncertainty quantification.
2.The integration of fuzzy memberships (combining possibility and necessity) offers a more reliable decision credibility measure than existing evidential approaches.
3.The DRF strategy effectively mitigates the adverse effects of conflicting views.

These claims are supported by comprehensive experimental results comparing FUML with both “untrusted” and “trusted” baselines on standard as well as deliberately corrupted (conflicting) test sets. The inclusion of ablation studies and additional analyses (in the supplementary materials) further substantiates the advantages claimed by the authors.

**Essential References Not Discussed:**

This paper has provides essential related works to understanding the key contributions of the work.

**Experimental Designs Or Analyses:**

The experimental design is thorough, incorporating experiments on eight public benchmarks with a mix of standard and conflicting instances. The authors report results over multiple random seeds to capture variability, and ablation studies are conducted to isolate the contribution of each component of FUML.

**Methods And Evaluation Criteria:**

The methodology is built upon a solid foundation of Fuzzy Set Theory. Modeling the classifier outputs as fuzzy memberships is an innovative approach that enables the joint estimation of possibility (likelihood) and necessity (exclusion of other classes). The loss function is carefully designed to ensure that the network learns to align its predictions with the ground truth while avoiding over-optimization issues in the unmatched categories.
Evaluation is conducted on eight diverse datasets using standard metrics (accuracy, precision, F-score) and by reporting improvements over baselines. Moreover, the study uses both normal and synthetically corrupted testing sets to assess the robustness of the proposed uncertainty estimation and fusion strategy.

**Other Comments Or Suggestions:**

1.It may be helpful to include a discussion on potential real-world applications where the robust uncertainty estimation could be critical, as well as scenarios where the method might struggle.
1.A more detailed error analysis could provide insights into the types of conflicting instances where FUML offers the greatest advantage.

**Other Strengths And Weaknesses:**

Strengths:
1.Novel use of Fuzzy Set Theory to model classifier outputs, resulting in a richer representation of decision credibility.
2.A well-motivated fusion strategy (DRF) that explicitly accounts for view-specific uncertainty and conflict.
3.Comprehensive experiments and ablation studies demonstrating both accuracy gains and improved uncertainty estimates.

Weaknesses:
1.Clearer explanations of the theoretical components and a more intuitive discussion of the loss function behavior are encouraged.
2.Details on computational complexity and hyperparameter sensitivity are limited.
3.Additional discussion on limitations and potential failure cases in real-world noisy environments would be useful.

**Questions For Authors:**

1. Why do you use $(1 - u_i^v)(1 - o_i^v)$? Would other combinations (e.g., weighted sum) improve performance?
2. How sensitive is the performance of FUML to the choice of the normalization and activation functions in mapping logits to fuzzy memberships? Could alternative functions affect the uncertainty estimation?
3. Can you provide further insight into the computational complexity of the DRF strategy compared to conventional fusion methods?
4. Could you discuss on the robustness of FUML in the presence of adversarial noise, and whether the uncertainty estimation mechanism is capable of flagging such instances effectively?

**Relation To Broader Scientific Literature:**

FUML is positioned well within the existing body of work on multi-view learning and uncertainty estimation. It builds upon and extends previous methods such as Evidential Deep Learning-based approaches (e.g., ETMC, TMC, ECML) and Dempster-Shafer theory, while introducing the use of fuzzy set theory, which is a perspective that has been underexplored in this context. The paper also clearly outlines how its contributions differentiate from prior works by addressing the limitation of underestimating uncertainty in conflicting instances.

**Theoretical Claims:**

The paper presents two theoretical propositions (Propositions 3.4 and 3.5) which state that fusing a clean view with a conflicting view increases the overall uncertainty, which is a desirable property for flagging unreliable decisions. The proofs and reasoning are convincing.

---

> ### Author Rebuttal · Authors · 2025-04-01
>
> We appreciate your valuable comments. Below is our point-by-point response.
>
> **Q1: Weaknesses (1)**
>
> **R1**: 1) In the next version, we will provide a clearer explanation of the theoretical components. 2) An intuitive discussion of the loss function is as follows: Directly aligning category credibility $(m^j_{ik} + 1 - \max\{m_{il}^j \mid l \neq k \})/2$ with the label $y^j_{ik}$ will lead to local optima. Specifically, when $y^j_{ik} = 0$ and $y^j_{il} = 0 (l \neq k)$, minimizing $(1 - \max \\{m_{il}^j \mid l \neq k \\} )$ drives $m_{il}^j$ toward 1 instead of 0, leading to incorrect optimization. To address this, the loss function (Eq.(4)) replaces $(m^j_{ik} + 1 - \max\\{m_{il}^j \mid l \neq k \\})/2$  with $(m^j_{ik} + 1 - m^j_{il})/2$  when $y^j_{ik} = 0$, ensuring correct optimization. We will clarify this in the next version.
>
> **Q2: Weaknesses (3)**
>
> **R2**:  Adverse weather conditions and data communication issues often introduce noise and misalignment in multi-view data. Existing EDL-based TMVC methods struggle with accurate classification under such conflicts and may underestimate classification uncertainty. In the next version, we will include this discussion in the Introduction Section.
>
> **Q3**: Other Comments Or Suggestions (1).
>
> **R3**: 1) In medical diagnosis, integrating multiple medical images (e.g., CT, MRI) and genetic data is crucial for disease classification. However, noise or pathological differences between imaging devices would cause inconsistencies, necessitating accurate uncertainty estimation about the classification results for doctors (e.g., ``classified as a malignant tumor, but MRI and CT conflict'') to assist in risk assessment. 2) Our FUML exhibits limited effectiveness in two-view data scenarios. These discussions will be included in the next version.
>
> **Q4: Other Comments Or Suggestions (2)**
>
> **R4**: Appendix Tables 7–9 show FUML achieving top or near-top performance in conflicting MVC across eight datasets with diverse noise and misalignment. Notably, FUML significantly outperforms other baselines on unaligned datasets. Overall, FUML demonstrates robustness in handling noisy and unaligned multi-view data, particularly in unaligned cases. This discussion will be incorporated into Appendix C.1 in the next version.
>
> **Q5: Questions For Authors (1)**
>
> **R5**: 1) Environmental factors often cause conflicts, which can mislead classification decisions. To mitigate this, we propose fusing only clear (low-uncertainty) and well-aligned (low-conflict) views. Consequently, we employ $(1−u_i^v)(1−o_i^v)$. 2) No, other combinations fail to improve performance. The table below proves this.
>
> |                             | Fashion         | LandUse         |
> | --------------------------- | --------------- | --------------- |
> | $0.5(1-u^v_i)+0.5(1-o^v_i)$ | $96.22\pm0.26$  | $39.36\pm5.82$  |
> | $0.8(1-u^v_i)+0.2(1-o^v_i)$ | $95.97\pm 0.31$ | $48.71\pm11.16$ |
> | $0.2(1-u^v_i)+0.8(1-o^v_i)$ | $96.30\pm 0.54$ | $59.26\pm 5.16$ |
> | Ours                        | $96.68\pm0.32$  | $69.14\pm2.43$  |
>
> **Q6: Questions For Authors (2)**
>
> **R6**: Regarding normalization, Appendix C.6 details the impact of the normalization parameter p on performance. The results show that FUML generally achieves optimal performance when p is between 2 and 5. As for activation functions, ReLU is essential for quantizing membership. Using alternative functions (exp or softplus) would yield outputs outside the $[0,1]$, violating fuzzy set theory's membership definition and precluding subsequent category credibility modeling and uncertainty estimation.
>
> **Q7: Weaknesses (2) & Questions For Authors (3)**
>
> **R7**:  Although the computational complexity is $O(V^2)$, the number of views in multi-view classification (MVC) is typically small ($V < 10$), making the additional computational overhead negligible.  Our experiments demonstrate that our method exhibits no significant increase in inference time compared to baselines, approximately by less than $8\\%$ (i.e., 0.0231s vs. 0.0215s).
>
> **Q8: Questions For Authors (4)**
>
> **R8**: 1)We add PGD adversarial noise attacks with different maximum perturbation magnitudes (eps) to the test set of the Fashion dataset. The results below show FUML's superior resistance to adversarial noise attacks. 2)We perform an OOD task on the Fashion dataset, using normal test sets as ID and PGD-attacked sets (eps=0.10) as OOD. Evaluated by FPR95 (lower is better), FUML achieved 0.68, outperforming ETMC and ECML (both 1.00).
>
> |            | 0               | 0.05            | 0.10           |
> | ---------- | --------------- | --------------- | -------------- |
> | PDF        | $98.95\pm0.19$  | $22.07\pm0.90$  | $13.54\pm0.89$ |
> | ECML       | $97.93\pm 0.35$ | $52.58\pm 0.51$ | $42.74\pm1.35$ |
> | FUML(Ours) | $98.96\pm0.25$  | $94.45\pm0.18$  | $93.40\pm0.19$ |

---

> > ### Comment · Reviewer_8eiq · 2025-04-04
> >
> > After a thorough review of the rebuttal, I am convinced that the authors have satisfactorily addressed all of my concerns. They have provided robust evidence and detailed experimental results that not only clarify the raised issues but also highlight the novel contribution of integrating fuzzy set theory for trusted classification. Furthermore, the authors' responses to the comments of other reviewers reinforce the strength and validity of their approach. The novel use of fuzzy memberships to capture decision credibility, alongside comprehensive ablation studies and comparative experiments, convincingly demonstrates both improved classification accuracy and enhanced uncertainty estimation. In light of these points, I fully support accepting this paper.

---

> > > ### Author Response · Authors · 2025-04-09
> > >
> > > Thank you very much for your support. We will further improve the quality of the final manuscript based on your constructive suggestions.

---

### Official Review · Reviewer_Ffiy · 2025-03-13

**Overall Recommendation:** 3

**Summary:**

This paper proposes a deep fuzzy multi-view learning method (FUML) to classify conflicting multi-view instances and precisely estimate intrinsic uncertainty. Specifically, FUML models logits as fuzzy memberships, employs Shannon's entropy to estimate uncertainty, and utilizes the cosine metric to measure the conflict degree between views. These factors are then combined to perform weighted fusion for multi-view classification.

**Claims And Evidence:**

Yes

**Essential References Not Discussed:**

The paper has included the main related works that are crucial for understanding the context.

**Experimental Designs Or Analyses:**

I have checked all the experiments in the experimental section.

**Methods And Evaluation Criteria:**

Yes, the proposed method and evaluation criteria make sense for the multi-view classification.

**Other Comments Or Suggestions:**

See Weaknesses.

**Other Strengths And Weaknesses:**

Strengths
1. The topic of this paper is interesting as it addresses conflict-aware multi-view fusion.
2. The writing of the paper is clear and well-structured.

After carefully reviewing this paper, I have the following concerns:

(1) Concerns about uncertainty estimation

This paper uses Shannon's entropy to estimate the uncertainty of multi-view results. However, entropy primarily measures the uniformity of category distributions, making it unreliable for distinguishing between hard-to-classify in-distribution (ID) samples and true out-of-distribution (OOD) samples, as both may exhibit high entropy. Additionally, deep neural networks often suffer from overconfidence, and without proper uncertainty calibration, OOD samples may be assigned artificially low entropy when misclassified with high confidence. It remains unclear why Shannon's entropy alone is used as an uncertainty measure for noisy or OOD detection tasks, as demonstrated in C.5.

(2) Concerns about conflict measurement

The paper uses cosine similarity to estimate the conflict degree among multiple views. However, this approach may overestimate conflicts in cases where two views share the same most probable class but differ in lower-ranked class distributions. This can lead to unnecessary penalization of views that are actually aligned in their final decision. Additionally, the method does not account for the varying importance of different categories, which may result in suboptimal weighting in the fusion process.

(3) Concerns about the fusion method

1. The current fusion method employs an exponential weighting mechanism based on the product of uncertainty and conflict degree. While this approach effectively suppresses highly uncertain and highly conflicting views, it may introduce a notable limitation: i.e., low uncertainty inherently diminishes the impact of conflict degree. This issue arises due to the multiplicative interaction between uncertainty and conflict, which implicitly assumes that both factors should jointly determine reliability in a symmetric manner.

2. This method finds it difficult to address conflicts when fusing only two views with the same uncertainty degree due to the symmetry of cosine similarity.

(4)  Concerns about computational complexity

This method uses Eq.(7) to measure the conflict degree, introducing a quadratic computational complexity ($O(V^2)$) for pairwise conflict computation, which raises scalability concerns for high-dimensional multi-view data.

(5)  Concerns about experiments

It is suggested to add an ablation study to analyze the effects of uncertainty and conflict separately in multi-view fusion. Furthermore, as this paper is evaluated on feature-based multi-view datasets, it is recommended to conduct experiments on multi-modal datasets, such as BRCA, LGG, and NYUD2, to enhance the evaluation. Moreover, it is recommended to compare some of the latest methods for conflicting view fusion, e.g., [1].

(6) Minor Issues

The paper should be carefully double-checked, as there are some errors. For example, it should be $m_{il}^v$ in Eq.(1), and $k$ should be italicized in Line 211.

[1] Navigating Conflicting Views: Harnessing Trust for Learning.

**Questions For Authors:**

Overall, considering the points raised in the Major Weaknesses section, I have assigned the current rating accordingly. I am open to further discussion with the authors if there are any misunderstandings in my review. I will make the final decision based on the authors' responses.

**Relation To Broader Scientific Literature:**

The paper is closely related to multi-view learning and uncertainty estimation. It makes improvement to conflicting multi-view fusion.

**Theoretical Claims:**

I have checked all Theoretical Claims.

---

> ### Author Rebuttal · Authors · 2025-04-01
>
> We sincerely appreciate your valuable feedback. Below are our point-to-point responses:
>
> **Q1: Concern 1**
>
> **R1**: In fact, we don't only use Shannon’s entropy for OOD detection. The uncertainty estimation mechanism of FUML includes the following steps: We first calculate the category credibility based on fuzzy membership. The category credibility contains possibility and necessity information, thereby alleviating overconfidence. Then we use Shannon’s entropy of category credibility to estimate uncertainty.  The closer the category credibility is to 0.5, the greater the entropy, indicating that the model is less able to make decisions. As can be seen in Figure 8, the uncertainty increases with the increase of noise intensity, which proves the effectiveness of our FUML for OOD detection.
>
> **Q2: Concern 2**
>
> **R2**: 1) In fact, rather than overestimating the conflict between two views, our approach correctly reflects subtle semantic differences. Specifically, the difference in lower-ranked class distributions represents the unique characteristics and information captured by each view. For instance, in Figure 1(a), despite both the RGB and text views being classified as “bathroom”, the RGB view includes additional semantics like “toilet paper” and “toilet”, which the text view lacks.  2) We are confused about your intentions. This paper considers multi-view fusion and doesn't take into account the long-tail distribution.
>
> **Q3: Concern 3**
>
> **R3**: 1) Our assumption that uncertainty and conflict should jointly determine fusion weights is well-founded: Either higher uncertainty or conflict will reduce its reliability, and it is reliable only if both are low. Specifically, high uncertainty reflects the inherent unreliability of predictions from the corresponding view, while high conflict indicates inconsistencies between decisions from different views, both of which lead to unreliable fusion outcomes. Thus, we use uncertainty and conflict together to compute the fusion weights, so as to more accurately reflect the reliability of the corresponding views. 2) When fusing two uncertainly identical and conflicting views, it is impossible to make a correct decision from a human perspective, and therefore our FUML can't either.
>
> **Q4: Concern 4**
>
> **R4**: Although the computational complexity is $O(V^2)$, the number of views is small (usually $V < 10$) in MVC, so the extra computational overhead is negligible. Our experiments indicate that our method exhibits no significant increase in inference time compared to baselines, with an approximate increase of $8\\%$ (i.e., 0.0231s vs. 0.0215s)
>
> **Q5: Concern 5**
>
> **R5**: 1) We conduct ablation experiments on the conflicting test sets. The classification accuracy below shows that removing $u_i^v$ or $o_i^v$ leads to performance degradation, indicating the effectiveness of considering both uncertainty and conflict.
>
> |         | AVG              | $1-u^v_i$        | $1-o^v_i$        | Ours             |
> | ------- | ---------------- | ---------------- | ---------------- | ---------------- |
> | Fashion | $96.15 \pm 0.22$ | $96.27 \pm 0.22$ | $96.43 \pm 0.32$ | $96.68 \pm 0.32$ |
> | LandUse | $67.71 \pm 2.30$ | $68.14 \pm 2.47$ | $68.50 \pm 2.02$ | $69.14 \pm 2.28$ |
>
> 2)Since LGG is unavailable, we substitute it with Prokaryotic[1]. The experimental results below demonstrate the superior performance of our FUML. On the NYUD2 dataset with Gaussian noise (mean=0, variance=5), FUML demonstrates superior performance, achieving a classification accuracy of $61.44\pm1.84$ compared to baseline methods (TMC: $59.12\pm1.98$, QMF: $60.32\pm2.63$, PDF: $61.83\pm1.78$).
>
> |             | PDF            | ECML           | CCML           | Ours           |
> | ----------- | -------------- | -------------- | -------------- | -------------- |
> | BRCA        | $82.40\pm1.27$ | $81.49\pm0.86$ | $79.66\pm1.86$ | $81.78\pm1.24$ |
> | Prokaryotic | $70.18\pm4.79$ | $55.05\pm5.23$ | $54.95\pm5.18$ | $73.15\pm4.90$ |
>
> 3)Due to the unavailable source code for "Navigating Conflicting Views: Harnessing Trust for Learning", we compared our FUML with another latest work, TUNED (AAAI 2025)[4], as shown blow, demonstrating the superiority of our method.
>
> |       | HW               | NUSOBJ           | Fashion          | Scene            | LandUse          |
> | ----- | ---------------- | ---------------- | ---------------- | ---------------- | ---------------- |
> | THNED | $96.75 \pm 0.55$ | $34.09 \pm 0.14$ | $86.99 \pm 0.75$ | $67.22 \pm 0.58$ | $46.64 \pm 2.10$ |
> | Ours  | $98.78 \pm 0.36$ | $47.08 \pm 0.32$ | $96.68 \pm 0.32$ | $72.71 \pm 1.75$ | $69.14 \pm 2.43$ |
>
> **Q6: Concern 6**
>
> **R6**: We will correct Eq.(1) and thoroughly proofread the paper for any remaining errors in the next version.
>
> [1]The landscape of microbial phenotypic traits and associated genes.
>
> [2]Trusted Unified Feature-Neighborhood Dynamics for Multi-View Classification.

---

> > ### Comment · Reviewer_Ffiy · 2025-04-05
> >
> > Thanks to the authors for their response. However, I believe my main concerns remain largely unaddressed.
> >
> > **(1) Concerns about uncertainty estimation**
> >
> > The authors mention that category credibility is based on fuzzy membership and incorporates possibility and necessity to mitigate overconfidence. This seems to refer to the loss in Eq. (4), which is only active during training. However, during inference, the model uses a deterministic pipeline—from logits to fuzzy memberships via $L_p$-norm and ReLU, then to category credibility and entropy-based uncertainty estimation—without any calibration techniques (e.g., temperature scaling).
> >
> > As a result, in out-of-distribution (OOD) cases—where Eq. (4) imposes no constraint during training due to the absence of label supervision—the model may still produce highly confident but incorrect outputs. For instance, a fuzzy membership like [0.8,0.1,0.1] leads to a category credibility of approximately [0.85,0.15,0.15], still reflecting strong overconfidence.
> >
> > Moreover, the uncertainty in FUML (Eq. 6) is computed as the average of binary entropy values across all category credibility scores. This treats all classes independently and equally, regardless of their semantic relevance. For example, even low credibility scores (e.g., 0.15) contribute significantly to total uncertainty. Thus, a totally correct prediction like [0.85,0.15,0.15], which clearly favors one class, still yields moderate uncertainty (≈0.61), which contradicts the intuitive expectation that such confident predictions should yield lower uncertainty.
> >
> > **(2) Concerns about conflict measurement**
> >
> > My concern is not whether views encode subtle semantic differences, but rather the risk of over-penalizing semantically aligned views. The current conflict metric, based on cosine similarity between class distribution vectors, does not account for top-1 prediction agreement.
> >
> > For example, consider the following four views:
> >
> > view1: [0.85, 0.10, 0.05] (Correct)
> >
> > view2: [0.60, 0.15, 0.25] (Correct)
> >
> > view3: [0.10, 0.85, 0.05] (Incorrect)
> >
> > view 4: [0.20, 0.70, 0.10] (Incorrect)
> >
> > The conflict scores are: $o_1 = 0.479$, $o_2 = 0.400$, $o_3 = 0.477$, and $o_4=0.372$. Despite View1 and View2 being correct, their conflicts are as high as (or higher than) the incorrect views, which may distort the fusion process by penalizing accurate views.
> >
> > While I acknowledge that the conflict metric is intended to capture distributional disagreement rather than prediction correctness, it is important to recognize that conflict values directly influence fusion weights in the current framework. This creates a mismatch between distributional difference and semantic trust, especially when a wrong prediction appears aligned at the distribution level. I recommend incorporating top-k class consistency or other semantic alignment mechanisms to improve the robustness of conflict estimation.
> >
> > **(3) Concerns about the fusion method**
> >
> > **a. Unreasonable multiplicative formulation in fusion weighting**
> >
> > I would like to emphasize that my concern is not about the idea of jointly considering uncertainty and conflict—which is entirely reasonable—but rather about the specific multiplicative formulation used to combine them. The exponential weighting function based on the product of uncertainty and conflict (Eq. 8) creates an imbalanced interaction: when either is small, the weight remains high, even if the other factor signals unreliability. For instance, a confident but semantically conflicting view may still be over-weighted, while a more moderate but consistent view may be under-weighted. This design fails to reflect true semantic reliability.
> >
> > **b. Inadequate handling of two conflicting views**
> >
> > Additionally, I understand the authors’ point that in some two-view cases, ambiguity is unavoidable even from a human perspective. However, this does not justify treating both views as equally reliable by default. In such cases, models may still benefit from leveraging subtle but actionable asymmetries, possibly including the confidence margin between top predictions, view-specific calibration behavior, or consistency with other samples in the batch (e.g., soft alignment). Unfortunately, the current fusion strategy—due to its strictly symmetric and multiplicative structure—has no mechanism to capture these signals. As a result, it is not that the model “can’t decide,” but rather that the design prevents it from trying. I encourage the authors to consider fusion strategies that allow for asymmetric cues or conditional weighting in the future, so that the model can still make meaningful distinctions even when uncertainty values are similar.

---

> > > ### Author Response · Authors · 2025-04-09
> > >
> > > We truly appreciate your valuable and constructive feedback. Below is our point-by-point response.
> > >
> > > **Q1: Concern 1**
> > >
> > > **R1**:
> > > 1) During inference, category credibility also incorporates the possibility and necessity measures, reducing overconfidence by providing uncertainty. Our FUML uses a deterministic pipeline but estimates uncertainty like EDL-based methods, and outperforms them in handling conflicting (Fig.3). Future work will explore calibration techniques to improve the accuracy of uncertainty.
> > >
> > > 2) FUML uses category credibility to reduce overconfidence. For example, an OOD sample with membership [0.8, 0.1, 0.1] yields an uncertainty of 0.55 using only membership, but derived category credibility [0.8, 0.15, 0.15] generates higher uncertainty (0.61). This indicates it reduces overconfidence for OOD samples by providing conservative uncertainty, as shown in C.5.
> > >
> > > 3) Eq.6 does not treat all classes independently; instead, it captures semantic relevance by reflecting how much a sample doesn‘t belong to alternative classes. This consideration of inter-class relationships, along with treating all category credibility equally, avoids underestimating potential risks by focusing solely on the main class, contributing to a more nuanced uncertainty estimation. The moderate uncertainty (0.61) of [0.85, 0.15, 0.15] prevents overconfidence. Conversely, more discriminative like [0.99, 0.0, 0.0], yields a low uncertainty (0.08), i.e., only reliable enough to generate low uncertainty.
> > >
> > > **Q2: Concern 2**
> > >
> > > **R2**:
> > > 1) The case you provided assumes access to labels for conflict evaluation, which is not practical at test time. Instead, our FUML computes conflict solely from the distribution of each view's memberships. In your example, it's the second class (not the first) that is dominant. Specifically, view3 and view4 both have high values in the second class, view1 and view3 share a similar level of discriminativeness, and view4 is more discriminative than view2 (0.70 vs. 0.60). So, the second class is more likely to be correct. Thus, the computed conflict scores are correct, where the conflicts of view3 and view4 are smaller than ones of view1 and view2, respectively, i.e., no clear over-penalization occurs in our FUML.
> > >
> > > 2) We acknowledge the potential mismatch between distributional differences and semantic trust. Our FUML addresses this by not relying solely on the conflict. View-specific uncertainty plays a complementary role in weighting decisions, enabling more robust fusion.
> > >
> > > 3) Based on your suggestion, we conducted experiments on the conflicting test set of Fashion. We compare top-k class consistency (retaining only top-k class probabilities and zeroing out the rest) with our FUML. Results below show our method outperforms top-K truncation, indicating that considering all classes enables more effective conflict measurement. Nonetheless, we greatly appreciate your suggestions and will explore semantic alignment mechanisms in the future.
> > >
> > > | top-1 | 96.45 |
> > > | ----- | ----- |
> > > | top-2 | 95.58 |
> > > | top-3 | 96.57 |
> > > | Ours  | 96.68 |
> > > **Q3: Concern 3**
> > >
> > > **R3**:
> > > a) We appreciate your perspective, yet we hold a different view. The purpose of using multiplication formulas is to avoid overestimating weights, which is also a highlight of this work. It ensures that either high uncertainty (u) or high conflict (o) is sufficient to reduce the influence of the corresponding view. For example:
> > >
> > > v1: u = 0.1, o= 0.1; (confident & consistent)
> > >
> > > v2: u = 0.1, o = 0.9; (confident & conflicting)
> > >
> > > v3: u= 0.5, o= 0.1. (moderate & consistent)
> > >
> > > As shown below, compared with a baseline ((1-u)+(1-o))/2, our FUML neither over-weights v2 nor under-weights v3, achieving a more balanced assessment.
> > >
> > > |          | v1   | v2   | v3   |
> > > | -------- | ---- | ---- | ---- |
> > > | baseline | 0.40 | 0.27 | 0.33 |
> > > | Ours     | 0.46 | 0.22 | 0.32 |
> > >
> > > b) In extreme cases where two conflicting views have the same uncertainty, our FUML does assign them equal weights. This doesn't mean it fails to make a decision. Instead, it offers a prediction with high decision uncertainty, which is appropriate under such ambiguity. Moreover, our FUML is scalable and can benefit from leveraging subtle but actionable asymmetries if relevant prior information is given. For example, our $(1-u^v) (1-o^v)$ can be extended to $q^v (1-u^v) (1-o^v)$ for asymmetric fusion, where $q^v$ represents a view-specific weight prior. We conduct experiments on the Fashion dataset, as shown below.
> > >
> > > | $q^1$; $q^2$; $q^3$ | Acc   |
> > > | ----- | ----- |
> > > | 0.6; 0.2; 0.2 | 96.61 |
> > > | 0.2; 0.6; 0.2 | 96.54 |
> > > | 0.2; 0.2; 0.6 | 96.44 |
> > > | Ours            | 96.68 |
> > > From the results, hardcoded priors don't yield performance gains. However, we agree with your insightful suggestion, such as confidence margins or cross-sample consistency, to conditionally weight views. This is a promising direction for future work, and we thank you again for the valuable feedback.

---

### Decision · Program_Chairs · 2025-05-01

**Decision:**

Accept (poster)

**Comment:**

This paper proposes a novel deep fuzzy multi-view learning method called FUML. This method leverages fuzzy set theory to model the outputs of a classification neural network as fuzzy memberships, incorporating both possibility and necessity measures to quantify category credibility, optimized by a tailored loss function. To further enhance uncertainty estimation, this paper proposes an entropy-based uncertainty estimation method leveraging category credibility, and develops a dual reliable multi-view fusion strategy that accounts for both view-specific uncertainty and inter-view conflict to mitigate the influence of conflicting views in multi-view fusion. Extensive experiments demonstrate that the proposed method achieves SOTA performance in terms of both accuracy and reliability.

After the rebuttal process, this paper finally receives 1 weak accept, 2 accept, 1 strong accept. All the reviewers reach an agreement that this paper can be accepted. So I recommend accepting this paper.